# Differential Privacy of Cross-Attention with Provable Guarantee

## Abstract

Cross-attention has become a fundamental module nowadays in many important artificial intelligence applications, e.g., retrieval-augmented generation (RAG), system prompt, guided stable diffusion, and many more. Ensuring cross-attention privacy is crucial and urgently needed because its key and value matrices may contain sensitive information about model providers and their users. In this work, we design a novel differential privacy (DP) data structure to address the privacy security of cross-attention with a theoretical guarantee. In detail, let $n$ be the input token length of system prompt/RAG data, $d$ be the feature dimension, $R$ be the maximum value of the query and key matrices, $R_w$ be the maximum value of the value matrix, and $r, s, \epsilon_s$ be parameters of polynomial kernel methods. Then, our data structure requires $\widetilde{O}(ndr^2)$ memory consumption with $\widetilde{O}(ndr^2)$ initialization time complexity and $\widetilde{O}(dr^2)$ query time complexity for a single token query. In addition, our data structure can guarantee that the process of answering user query satisfies $(\epsilon, \delta)$-DP with $\widetilde{O}((1 - \epsilon_s)^{-1} n^{-1} \epsilon^{-1} R^{2s} R_w r^2)$ additive error and $2\epsilon_s/(1 - \epsilon_s)$ relative error between our output and the true answer. Furthermore, our result is robust to adaptive queries in which users can intentionally attack the cross-attention system. To our knowledge, this is the first work to provide DP for cross-attention and is promising to inspire more privacy algorithm design in large generative models (LGMs).

## 1 Introduction

The development of Artificial Intelligence (AI) has four stages: (1) prediction AI, e.g., ResNet (He et al., 2016) in image classification; (2) generation AI, e.g., ChatGPT (Achiam et al., 2023) in language generation; (3) autonomous agent AI, Voyager (Wang et al., 2023a) autonomously plays Minecraft game (Fan et al., 2022); (4) Artificial Generalization Intelligence (AGI). Humans have made rapid progress in generative AI, and we are excitingly heading to the third stage, the era of AI agent (Liu et al., 2023). One prevalent application of AI agents is customized large generative models (LGMs) agents (OpenAI, 2024a), e.g., AgentGPT (GitHub, 2024a), SuperAGI (GitHub, 2024d), MetaGPT (Hong et al., 2024b;a), GPT Researcher (GitHub, 2024c) and many so on. In particular, recently, Apple Inc. introduced Apple Intelligence (Apple, 2024), signaling the integration of LGMs into physical devices. This innovation allows devices to use personal information for real-life assistance, such as entering passport numbers when booking flights or informing users of their latest meetings. With increased AI capabilities, privacy concerns become significant, as the more personal information devices handle, the greater the potential privacy risks.

One fundamental technique used in LGMs is cross-attention (Vaswani et al., 2017), which is an essential module in retrieval-augmented generation (RAG) (Lewis et al., 2020), system prompt, guided stable diffusion, and many so on. In RAG, to be more professional, the LGMs answer user input queries by using a domain-specific database under cross-attention, which may contain specific privacy data and knowledge so that the LGMs gain additional power. For system prompts, based on cross-attention, some customized long prompts, e.g., user information or concrete rules, are concatenated before user input to follow human instructions better, which are commonly used in ChatGPT (GitHub, 2024b), Claude3 (Anthropic, 2024) and other commercial LGMs.

Consequently, protecting the privacy of domain-specific data in RAG or system prompts is crucial as they contain sensitive information about users and companies. These data and prompts are the

core assets of many start-ups. However, these data and prompts can be easily recovered (Li et al., 2023b), jailbroken (Jin et al., 2024), and released (Li et al., 2023a) by user adversarial attack (Yu et al., 2024), e.g., there are 1700 tokens in ChatGPT system prompts (Patel, 2024). These findings highlight the critical importance of robust privacy protections in LGMs, making privacy not just essential but an urgent issue that demands immediate attention.

To fundamentally preserve cross-attention privacy, we borrow the powerful tools from differential privacy (DP) (Dwork et al., 2006), which provides measurable privacy and combines with statistical machine learning seamlessly (Ponomareva et al., 2023). Thus, in this work, we would like to ask and answer the following question,

*How can we use differential privacy to protect the security of cross-attention in LGMs?*

Our work demonstrates that the Softmax cross-attention computation is equivalent to computing the weighted distance problem.

**Definition 1.1** (Softmax cross-attention). *Let $n$ and $m$ be the token length of the data and input query, respectively. Let $d$ be the feature dimension. Given fixed key matrix $K \in [0, R]^{n \times d}$ and fixed value matrix $V \in [-R_w, R_w]^{n \times d}$, $R_w \geq 1$, for any input query matrix $Q \in [0, R]^{m \times d}$, the goal of the Softmax Cross-Attention Computation is to get the matrix $\mathrm{Attn}(Q, K, V) \in \mathbb{R}^{m \times d}$, which is*

$$\mathrm{Attn}(Q, K, V) := D^{-1} A V,$$

*where $A \in \mathbb{R}^{m \times n}$ satisfies $A_{i,j} := \exp(\langle Q_i, K_j \rangle / d)$ for any $i \in [m], j \in [n]$ ($Q_i$ and $K_j$ denote the $i$-th and $j$-th rows of $Q$ and $K$, respectively) and $D := \mathrm{diag}(A \mathbf{1}_n) \in \mathbb{R}^{m \times m}$ is a diagonal matrix.*

Note that $\mathsf{Softmax}(QK^\top / d) = D^{-1} A \in \mathbb{R}^{m \times n}$ in Definition 1.1, which is the standard function used in transformers, and usually, we call it as attention matrix. Our main theorem, presented below, provides a robust solution of cross-attention, ensuring privacy and accuracy guarantees.

**Theorem 1.2** (Main result; Informal version of Theorem 3.1). *Let $Q, K, V, \mathrm{Attn}$ be defined in Definition 1.1. Let $p_f$ be the probability of failure parameter. Let $r, s, \epsilon_s$ be the parameters of the polynomial kernel methods (Lemma H.6). Then, our Algorithm 1 requires $\widetilde{O}(ndr^2)$ memory with $\widetilde{O}(ndr^2)$ initialization time and $\widetilde{O}(dr^2)$ query time, such that with probability $1 - p_f$, the output process of cross-attention satisfies $(\epsilon, \delta)$-DP and is robust to adaptive query with relative error $2\epsilon_s / (1 - \epsilon_s)$ and additive error $\widetilde{O}((1 - \epsilon_s)^{-1} n^{-1} \epsilon^{-1} R^{2s} R_w r^2)$.*

Our main technique in Theorem 1.2 ensures that cross-attention is differentially private by using the polynomial kernel approximation method and transforming it into a weighted distance problem. We then solve the problem by summing over weighted distances (depending on the value embedding) between the query embedding and the key embedding. We build a data structure for weighted Softmax queries in Section 4.3, and we extend this data structure to handle adaptive queries using the $\epsilon_0$-net/metric entropy argument in Section 4.4. Furthermore, our additive error decreases as the input token length grows, diminishing to zero.

Our contributions are as follows:

- We demonstrate that cross-attention computations are equivalent to the weighted distance problem (Section 3).
- We design a novel algorithm (Algorithm 3) that privately answers weighted Softmax queries with high probability and a concrete accuracy bound.
- Our algorithm (Algorithm 1) handles multiple cross-attention queries and is robust against adaptive query attacks (Theorem 3.1), meaning that potential attackers cannot intentionally extract information of system prompts/RAG data.

To our knowledge, this is the first work to utilize DP to protect prompts in LGMs with theoretically provable guarantees. While some have explored protecting user/system prompts with DP (Edemacu & Wu, 2024; Mai et al., 2023), they are primarily empirical and lack theoretical guarantees. Additionally, many others are working on protecting private datasets by applying DP to the fine-tuning stage of LGMs (Behnia et al., 2022; Singh et al., 2024; Liu et al., 2024b; Yu et al., 2021; Li et al., 2021; Shi et al., 2022a), which diverges from our work. The strength of DP lies in its strong, unambiguous, and concrete definition of privacy, enabling algorithm designs with provable privacy and

accuracy analysis. Therefore, we believe that the theoretical aspects of DP applications in LGMs remain a highly impactful direction, and we aim to pave the way for further exploration in this area.

### 1.1 RELATED WORK

**Differential Privacy in Data Structure and Attention.** Differential privacy (DP) is a flourishing and powerful technique that has enormous applications in the topic of private machine learning. In the era of Large Generative Models (LGMs), there are three primary approaches to ensuring privacy: (1) during the pre-training stage: to protect training data (Abadi et al., 2016; Ponomareva et al., 2023), (2) during the adaptation stage: to protect target data (Behnia et al., 2022; Singh et al., 2024; Liu et al., 2024b; Yu et al., 2021; Li et al., 2021; Shi et al., 2022a; Huang et al., 2024), (3) during the inference stage: to protect user/system prompts (Edemacu & Wu, 2024) and RAG data (Lewis et al., 2020). To protect training data, DP-SGD (Abadi et al., 2016) uses DP optimizer to ensure data privacy, severing as the traditional baseline method. Recently, numerous works have aimed to improve this method by integrating DP in both the pre-training and fine-tuning stages of LGMs (Yu et al., 2021; Li et al., 2021; Golatkar et al., 2022; Behnia et al., 2022; Shi et al., 2022a; Mattern et al., 2022; Singh et al., 2024; Zheng et al., 2024; Liu et al., 2024b). However, DP-SGD confines differential privacy to the optimizer. In contrast, we propose a novel approach that integrates DP directly into the attention mechanism, supported by strong theoretical analysis and guarantees. Given the resource-intensive nature of training LGMs, our technique offers a practical alternative for models trained with standard SGD, which lack inherent privacy guarantees. In such cases, applying DP-SGD would require retraining the models, which is computationally expensive, whereas our method avoids this additional cost.

To protect user/system prompts, Edemacu & Wu (2024) provides a survey on both DP and non-DP methods. In the use of LGMs, prompting methods almost become a standard way for inference (Schulhoff et al., 2024). Given the billions of prompt interactions daily, ensuring privacy is essential (Mai et al., 2023). We refer readers to Appendix A for more related works.

**Roadmap.** In Section 2, we present the preliminary of differential privacy (DP) and cross-attention. In Section 3, we present the main result of our cross-attention theorem (Theorem 3.1). In Section 4, we outline the main results of our algorithms. In Section 5, we discuss DP-related topics and potential extensions. In Section 6, we conclude our paper.

## 2 PRELIMINARY

In this section, we give the preliminary of differential privacy (DP) and cross-attention. In Section 2.1, we describe the notations. In Section 2.2, we give definitions related to DP.

### 2.1 NOTATIONS

We use $\Pr[]$ to denote the probability. We use $\mathbb{E}[]$ to denote the expectation. We use $\mathrm{Var}[]$ to denote the variance. For two vectors $x \in \mathbb{R}^d$ and $y \in \mathbb{R}^d$, we use $\langle x, y \rangle$ to denote the inner product between $x, y$, i.e., $\langle x, y \rangle = \sum_{i=1}^d x_i y_i$. We use $X \subset \mathbb{R}^d$ and $|X| = n$ to mean the same thing as $X \in \mathbb{R}^{n \times d}$. Also, we denote $x_i^\top$ as the $i$-th row of $X$. We use $x_{i,j}$ to denote the $j$-th coordinate of $x_i \in \mathbb{R}^n$. We use $\mathbf{1}_n$ to denote a length-$n$ vector where all the entries are ones. We use $\|x\|_p$ to denote the $\ell_p$ norm of a vector $x \in \mathbb{R}^n$, i.e., $\|x\|_1 := \sum_{i=1}^n |x_i|$, $\|x\|_2 := (\sum_{i=1}^n x_i^2)^{1/2}$, and $\|x\|_\infty := \max_{i \in [n]} |x_i|$. We denote polynomial time complexity with respect to $n$ as $\mathrm{poly}(n)$. For a function $f$, we use $\widetilde{O}(f)$ to represent $f$ multiplied by a polylogarithmic factor, i.e., $f \cdot \mathrm{poly}(\log f)$. This notation, known as soft-$O$ or tilde notation, simplifies expressions by omitting logarithmic factors, focusing on the dominant term's growth rate.

### 2.2 DIFFERENTIAL PRIVACY DEFINITIONS

In this section, we give several definitions related to differential privacy (DP). We refer the reader to Dwork & Roth (2014) for more background and details on DP.

**Definition 2.1** (Neighboring dataset)**.** *Two datasets $X, X' \in [0, R]^{n \times d}$ are neighboring if they differ in exactly one row, i.e., there exists $i \in [n]$ such that $X_{i,*} \neq X'_{i,*}$ and $X_{j,*} = X'_{j,*}$ for all $j \neq i$.*

**Definition 2.2** (Sensitivity). *The sensitivity of a function* $f : \mathbb{R}^{n \times d} \to \mathbb{R}^{n \times d'}$ *is:* $\Delta :=$ $\max_{X, X' \in \mathbb{R}^{n \times d}} \|f(X) - f(X')\|_1$, *where* $X, X'$ *are neighboring datasets and* $\|\cdot\|_1$ *is the entry-wise* $\ell_1$*-norm.*

**Definition 2.3** (($\epsilon, \delta$)-DP). *For* $\epsilon > 0, \delta \geq 0$, *a randomized algorithm* $\mathcal{A}$ *is* ($\epsilon, \delta$)*-DP, if for all* $\mathcal{S} \subseteq \mathrm{Range}(\mathcal{A})$ *and for all neighboring datasets* $X, X'$ ~~such that~~ $\|X - X'\|_1 \leq 1$:

$$\Pr[\mathcal{A}(X) \in \mathcal{S}] \leq \exp(\epsilon) \Pr[\mathcal{A}(X') \in \mathcal{S}] + \delta.$$

*When* $\delta = 0$, *the algorithm is said to have pure differential privacy.*

We mainly use the truncated Laplace mechanism, which has the following definitions.

**Definition 2.4** (Truncated Laplace distribution). *We use* $\mathrm{TLap}(\Delta, \epsilon, \delta)$ *to denote the Truncated Laplace distribution with pdf proportional to* $\exp(-\epsilon|z|/\Delta)$ *on the region* $[-B, B]$, *where* $B = \frac{\Delta}{\epsilon} \cdot \log(1 + \frac{\exp(\epsilon) - 1}{2\delta})$.

**Fact 2.5** (Theorem 3 in Geng et al. (2020)). *Let* $z$ *denote a* $\mathrm{TLap}(\Delta, \epsilon, \delta)$ *random variable. Then we have* $\mathbb{E}[z] = 0$, *and*

$$\mathrm{Var}[z] = \frac{2\Delta^2}{\epsilon^2}(1 - \delta \cdot \frac{\log^2(1 + \frac{e^\epsilon - 1}{2\delta}) + 2\log(1 + \frac{e^\epsilon - 1}{2\delta})}{e^\epsilon - 1}).$$

*Furthermore, if* $\delta = 0$, *we have* $\mathrm{Var}[z] = 2\Delta^2/\epsilon^2$, *meaning truncated Laplacian mechanism will be reduced to the standard Laplacian mechanism.*

**Lemma 2.6** (Laplace mechanism, (Dwork & Roth, 2014; Geng et al., 2020), see Lemma 2.2 in Andoni et al. (2023)). *Given a numeric function* $f$ *that takes a dataset* $X$ *as the input, and has sensitivity* $\Delta$, *the mechanism that outputs* $f(X) + z$ *where* $z \sim \mathrm{Lap}(\Delta/\epsilon)$ *is* ($\epsilon, 0$)*-DP. In addition, if* $\epsilon, \delta \in (0, 0.5)$, $f(X) + z$, *where* $z \sim \mathrm{TLap}(\Delta, \epsilon, \delta)$ *is* ($\epsilon, \delta$)*-DP. Moreover, the truncated Laplace mechanism is always accuracy up to error* $B$.

---

**Algorithm 1** DP cross-attention algorithm

---
1: **datastrucutre** DPCROSSATTENTION $\qquad\qquad\qquad\qquad\qquad\qquad$ ▷ Theorem 3.1
2: **members**
3: $\quad$ $\mathcal{D}_0, \mathcal{D}_1, \ldots, \mathcal{D}_d$ : DPTREESOFTMAXADAPTIVE $\qquad\qquad\qquad$ ▷ Algorithm 7
4: **end members**
5: **procedure** INIT($K \in [0, R]^{n \times d}$, $V \in [-R_w, R_w]^{n \times d}$, $\epsilon \in (0, 1)$, $\delta \in (0, 1)$, $\delta' \in (0, 1)$, $c \in (0, 0.1), \epsilon_s \in (0, 0.1), p_f \in (0, 0.01)$) $\qquad\qquad\qquad\qquad\qquad$ ▷ $n = |K|$
6: $\quad$ **for** $k = 1 \to d$ **do**
7: $\qquad$ $\mathcal{D}_k$.INIT($K, n, V_{:,k}, \epsilon/2, \delta/2, \delta'/2, c, \epsilon_s, p_f$) $\qquad\qquad\qquad$ ▷ Compute $AV$
8: $\quad$ **end for**
9: $\quad$ $\mathcal{D}_0$.INIT($K, n, \mathbf{1}_n, \epsilon/2, \delta/2, \delta'/2, c, \epsilon_s, p_f$) $\qquad\qquad\qquad\qquad$ ▷ Compute $D$
10: **end procedure**
11: **procedure** QUERY($Q_i \in [0, R]^d$)
12: $\quad$ $O \leftarrow 0^d$
13: $\quad$ $D \leftarrow \mathcal{D}_0$.DISTANCEQUERY($Q_i$)
14: $\quad$ **for** $k = 1 \to d$ **do**
15: $\qquad$ $O_k \leftarrow D^{-1} \cdot \mathcal{D}_k$.DISTANCEQUERY($Q_i$)
16: $\quad$ **end for**
17: $\quad$ **return** $O$
18: **end procedure**
19: **end datastrucutre**

---

## 3 MAIN RESULTS: CROSS-ATTENTION

In this section, we show our main result for cross-attention. Theorem 3.1 states that we can ensure the entire cross-attention module satisfies DP and is robust to adaptive queries. Our high-level idea is based on the similarity between weighted distance problem and cross-attention. For a typical weighted distance problem, we define the following: Let $w \in \mathbb{R}^n$ be the weights, $X \in \mathbb{R}^{n \times d}$ be the

data matrix, where $x_i^\top$ is the $i$-th row of $X$ for $i \in [n]$, and let $y \in \mathbb{R}^d$ be the query. Suppose we need to answer $\ell_1$ distance query. We have

$$\sum_{i \in [n]} \underbrace{w_i}_{\text{weight}} \| \underbrace{y}_{\text{query}} - \underbrace{x_i}_{\text{data}} \|_1.$$

Now we introduce cross-attention. Let $Q, K, V, \mathrm{Attn}$ be defined in Definition 1.1. In a standard cross-attention process, $K$ and $V$ are accessible before inference, while the user input $Q$ becomes available only when the user provides it. Here, $K$ and $V$ represent values stored in memory or disks and are considered private assets protected within the model, whereas $Q$ is treated as public.

For the cross-attention mechanism $\mathrm{Attn}$ (Definition 1.1), we aim to ensure that the matrix $AV$ satisfies DP guarantee. Let $A_{i,j} = \exp(\langle Q_i, K_j \rangle / d)$ for $i \in [m], j \in [n]$. Let $V_{j,k} \in \mathbb{R}$ be the $(j, k)$-th entry of $V$, for $j \in [n], k \in [d]$. ~~Let $D = \mathrm{diag}(A\mathbf{1}_n)$, acting as a normalizing factor that aggregates all the information. We store both $K$ and its corresponding noises. For computing $AV$, we use the perturbed $K$, whereas for computing $D$, we rely on the original, unperturbed $K$. By post-processing property (Fact B.7), to ensure that the forward output $\mathrm{Attn}(Q, K, V) = D^{-1}AV$ (Definition 1.1) satisfies DP, we only need to ensure the DP of its component $AV$.~~

The $(i, k)$-th entry of $AV$ for each $i \in [m], k \in [d]$ is computed by

$$(AV)_{i,k} = \sum_{j=1}^{n} \underbrace{V_{j,k}}_{\text{weight}} \exp(\langle \underbrace{Q_i}_{\text{query}}, \underbrace{K_j}_{\text{data}} \rangle / d), \tag{1}$$

which can be viewed as a weighted Softmax problem, where $V$ provides the weights, $Q$ is the query, and $K$ is the dataset. Thus, we choose to add noise to $K$ and $V$ based on the similarity between the weighted distance problem and cross-attention. Furthermore, we find that we can only handle one column of $V$, i.e., $V_{*,k} \in \mathbb{R}^n$, in a single data structure. Therefore, we need to initialize a total of $d$ different data structures, each with weights $V_{*,k}$ for $k \in [d]$. For computing $D$, we treat $V = \mathbf{1}_n$, which can be interpreted as an weighted Softmax problem with weight $\mathbf{1}_n$.

Here, we present our main result below.

**Theorem 3.1** (Softmax cross-attention, informal version of Theorem H.11). *Let $Q, K, V, \mathrm{Attn}$ be defined in Definition 1.1. Assume the input context length $n$ is large enough. Let $p_f$ be the probability of failure parameter. Let $r, s, \epsilon_s$ be parameters of polynomial kernel methods (Lemma H.6). Let $\Gamma_{R,s} := \max_{j \in [s]} \frac{R^j}{\sqrt{j!}}$ (Definition H.3). Let $l = O(r \log(dR/(\epsilon_s p_f)))$. There is a data structure* DPTREECROSSATTENTION *(Algorithm 1) that uses $O(lnrd)$ spaces to ensure cross-attention DP and supports the following operations:*

- INIT$(K, V, \epsilon \in (0, 1), \delta \in (0, 1), \delta' \in (0, 1), c \in (0, 0.1), \epsilon_s \in (0, 0.1), p_f \in (0, 0.01))$ *(Algorithm 1). It takes $O(lnrd)$ time to initialize.*

- *At query time, for user input $Q$, we process one token at a time by passing the $i$-th row of $Q$, denoted $Q_i \in [0, R]^d$, to* QUERY$(Q_i)$ *(Algorithm 1) for each $i \in [m]$. It takes $O(ldr \log n)$ time to output an entry $z$ in $\mathrm{Attn}(Q, K, V)$ such that*
  - *the process of output $z$ satisfies $(\epsilon, \delta + \delta')$-DP,*
  - *the process of output $z$ has relative error $2\epsilon_s/(1 - \epsilon_s)$ and additive error $O((1 - \epsilon_s)^{-1}n^{-1}\epsilon^{-1}l\Gamma_{R,s}^2 R_w r \sqrt{\log(l/\delta')} \cdot \log^{3/2} n)$,*
  - *it holds with probability $1 - p_f$ (where $p_f$ is used in $l$),*
  - *it is robust to adaptive query.*

**Remark 3.2.** *Notice in Theorem 3.1 that we ensure the process of computing each entry is $(\epsilon, \delta + \delta')$-DP. To guarantee that the overall output vector of length $d$ is DP, we initialize each $\mathcal{D}_i$ for $i \in \{0, 1, 2, \ldots, d\}$ with parameters scaled from $\epsilon/2, \delta/2, \delta'/2$ to $\epsilon/(d + 1), \delta/(d + 1), \delta'/(d + 1)$. Then, by the basic composition property (Fact B.8), the output vector is $(\epsilon, \delta + \delta')$-DP, with the additive error increasing by a factor of $\widetilde{O}(d)$.*

In Theorem 3.1, we use our DPTREECROSSATTENTION (Algorithm 1) and guarantee that, for each query token of cross-attention, the output process satisfies $(\epsilon, \delta + \delta')$-DP with $2\epsilon_s/(1 - \epsilon_s)$ relative

error and $O((1 - \epsilon_s)^{-1}n^{-1}\epsilon^{-1}l\Gamma_{R,s}^2 R_w r\sqrt{\log(l/\delta')} \cdot \log^{3/2} n)$ additive error, and $O(ldr \log n)$ running time under adaptive query. More specifically, the algorithm creates $d + 1$ DPTREESOFT-MAXADAPTIVE (Algorithm 7) data structures, each requiring $O(lnr)$ memory consumption and $O(lnr)$ initialization time. Notably, our additive error is inversely proportional to $n$, meaning that as the input token length increases, the additive error approaches zero. ~~This is achieved by the normalizing matrix $D$ (Definition 1.1).~~ We refer the reader to Section H for proof details.

Thus, our algorithm theoretically protects system prompts/RAG data in cross-attention as discussed in Section 1. In Section 4, we provide a detailed technical overview, and in Section 5, we will present self-attention and DP-related discussion.

---

**Algorithm 2** DPTree initialization and query

1: **datastructure** DPTREE                                          ▷ Theorem C.1
2:   **members**
3:       $c : \mathbb{R}^{2n-1}$
4:   **end members**
5:   **procedure** INIT($a \in \mathbb{R}^n, n \in \mathbb{N}_+, \Delta \in \mathbb{R}, \epsilon \in (0,1), \delta \in (0,1)$)      ▷ Lemma C.4, Lemma C.3
6:       $b[n, 2n-1] \leftarrow a$
7:       **for** $i = n \rightarrow 2n - 1$ **do**
8:           $c[i] \leftarrow b[i] + \text{TLap}(\Delta, \epsilon/\log n, \delta/\log n)$
9:       **end for**
10:      **for** $i = (\log n) \rightarrow 1$ **do**
11:         **for** $j = 1 \rightarrow 2^{i-1}$ **do**
12:            $k \leftarrow 2^{i-1} + j - 1$
13:            $b[k] \leftarrow b[2k] + b[2k+1]$
14:            $c[k] \leftarrow b[k] + \text{TLap}(\Delta, \epsilon/\log n, \delta/\log n)$
15:         **end for**
16:      **end for**
17: **end procedure**
18: **procedure** QUERY($y \in [0, R]$)
19:      $c_{\text{left}}, c_{\text{right}} \leftarrow 0, 0$
20:      **for** $i = 1 \rightarrow \log n$ **do**
21:        Let node $j \in [2^i]$ of layer $i$ denotes the integer such that $y \in [(j-1)R/2^i, jR/2^i)$
22:        **if** $j$ is even **then**               ▷ Node $j$ is the right child of its parent
23:           $c_{\text{left}} \leftarrow c_{\text{left}} + c[2^i + j - 2]$      ▷ Add the value of left sibling node
24:        **else**                   ▷ Node $j$ is the left child of its parent
25:           $c_{\text{right}} \leftarrow c_{\text{right}} + c[2^i + j]$       ▷ Add the value of right sibling node
26:        **end if**
27:      **end for**
28:      **return** $c_{\text{left}}, c_{\text{right}}$
29: **end procedure**
30: **end datastructure**

---

## 4   KEY DATA STRUCTURE: DPTREE

This section provides our key data structures: DPTREE (Algorithm 2), DPTREEDISTANCE (Algorithm 4 and 5), DPTREEHIGHDIM (Algorithm 6), DPTREESOFTMAX (Algorithm 3), and DP-TREESOFTMAXADAPTIVE (Algorithm 7).

In Section 4.1, we provide our high-level proof insights. In Section 4.2, we give our basic building block algorithms DPTREE, DPTREEDISTANCE and DPTREEHIGHDIM. In Section 4.3, we present our DPTREESOFTMAX algorithm that solves the weighted Softmax problem. In Section 4.4, we present our DPTREESOFTMAXADAPTIVE algorithm that enables DPTREESOFTMAX to handle adaptive query problem.

### 4.1 TECHNIQUE OVERVIEW

Notice that Eq. (1) is not a typical distance measure like $\ell_1$ or $\ell_2$, but by using polynomial kernel method techniques, we transform it into a distance measure. Alman & Song (2023) states that the exponential inner product can be approximated by polynomial kernel function $P(\cdot) : \mathbb{R}^d \to \mathbb{R}^r$, i.e., $P(x)^\top P(y) \approx \exp(x^\top y/d)$ for two vector $x, y \in \mathbb{R}^d$, with a relative error. Then, by the Law of Cosines, we transform the inner product of polynomial kernel functions into a distance measure, i.e.,

$$2P(x)^\top P(y) = -\|P(x) - P(y)\|_2^2 + \|P(x)\|_2^2 + \|P(y)\|_2^2. \tag{2}$$

After transforming Eq. (1) into a distance measure, we design the DPTREE series data structures to provide cross-attention DP guarantee.

In summary, we first design the data structure DPTREE (Algorithm 2) that builds a binary segment tree with truncated Laplace noise added in the nodes to ensure DP guarantee. Then, based on this data structure, we design DPTREEDISTANCE (Algorithm 4 and 5) to answer one dimensional weighted $\ell_p^p$ distance queries $\sum_{i=1}^n w_i \cdot |y - x_i|^p$. We further decompose high dimensional $\ell_p^p$ distance problem into one dimensional $\ell_p^p$ distance problems using

$$\sum_{i=1}^n w_i \cdot \|y - x_i\|_p^p = \sum_{k=1}^d \sum_{i=1}^n w_i \cdot |y_k - x_{i,k}|^p. \tag{3}$$

Based on this decomposition, we design DPTREEHIGHDIM (Algorithm 6) which is capable of answering high dimension queries. Then, using Eq. (2) and DPTREEHIGHDIM, we design DP-TREESOFTMAX (Algorithm 3) to answer Softmax queries. By building multiple copies of this data structure, we boost the success probability such that it can answer any query (including adaptive query) with an additive error, establishing the final data structure DPTREECROSSATTENTION (Algorithm 1).

### 4.2 DPTREE, DPTREEDISTANCE, AND DPTREEHIGHDIM

The unweighted distance query has been explored in prior works (Huang & Roth, 2014; Backurs et al., 2024; Liu et al., 2024a). Specifically, Huang & Roth (2014) leverages online learning techniques to approximate the sum of distances, while Backurs et al. (2024) introduces a DP data structure based on a node-contaminated balanced binary tree. Furthermore, Liu et al. (2024a) presents a new data representation in tree nodes, where each node stores the sum of distances from one point to multiple points. In contrast, we focus on the weighted distance query, generalizing their results.

We design a basic data structure DPTREE (Algorithm 2) that answers summation queries by a summation segment tree with truncated Laplace noise (Definition 2.4). The algorithm first builds a binary summation tree in an array and then adds truncated Laplace noises to each node. During a query, the algorithm traverses each layer of the binary structure based on the input $y$, aggregating values from sibling nodes by accessing at most $O(\log n)$ nodes along the path. It then returns the accumulated left and right sums as the query result (Algorithm 2). See more details in Section C.

We then design DPTREEDISTANCE, a one-dimensional weighted $\ell_p^p$ distance data structure detailed in Algorithm 4 and 5. Initialization involves assigning each data point to the nearest bin and aggregating their weighted polynomial terms into multiple arrays (illustrated in Figure 1), which are then used to initialize several instances of our DPTREE. At query time, the algorithm retrieves aggregated weights from each DPTREE corresponding to the query point and combines them using binomial coefficients and distance powers to compute the one-dimensional weighted $\ell_p^p$ distance. Guided by Eq. (3), we design DPTREEHIGHDIM (Algorithm 6), which extends DPTREEDISTANCE to higher dimension by constructing independent data structures for each coordinate. See details in Section E and F.

### 4.3 SOFTMAX ACTIVATION

In this section, we present DPTREESOFTMAX (Algorithm 3) that answers the weighted Softmax query (Definition 4.1) and is further used to design DP cross-attention. First, we introduce the definition of weighted Softmax query, an abstraction for the problem described in Eq. (1).

---

**Algorithm 3** Softmax query

---

1: **datastrucutre** DPTREESOFTMAX            ▷ Theorem 4.2
2: **members**
3:    $\mathcal{D}_0, \mathcal{D}_1, \ldots, \mathcal{D}_r$ : DPTREEDISTANCE         ▷ Algorithm 4, Theorem E.1
4:    $P : [0, \Gamma_{R,s}]^{n \times r}$       ▷ Definition H.3 for $\Gamma_{R,s}$, Eq. (9) for $s$, Eq. (10) for $r$
5:    $w : [-R_w, R_w]^n$
6:    $P_{wx}, s_w, \epsilon_s : \mathbb{R}$
7: **end members**
8: **procedure** INIT($X \subset [0, R]^d$, $n \in \mathbb{N}_+$, $w \in [-R_w, R_w]^n$, $\epsilon \in (0, 1)$, $\delta \in (0, 1)$, $\delta' \in (0, 1)$,
   $c \in (0, 0.1)$, $\epsilon_s \in (0, 0.1)$)            ▷ Lemma H.6
9:    $\epsilon_s$, $w$, $P$, $P_{wx}$, $s_w \leftarrow \epsilon_s$, $w$, $0^{n \times r}$, $0$, $0$
10:    **for** $j = 1 \to n$ **do**
11:      Compute $P(x_j)$         ▷ Polynomial kernel function $P(\cdot)$, Lemma H.5
12:      ~~Compute $w_j \|P(x_j)\|_2^2$~~
13:      $P_{wx} \leftarrow P_{wx} + w_j \|P(x_j)\|_2^2$
14:      $s_w \leftarrow s_w + w_j$
15:      $P_{j,:} \leftarrow P(x_j)$
16:    **end for**
17:    **for** $i = 1 \to r$ **do**
18:      $\mathcal{D}_i$.INIT($P_{:,i}$, $n$, $w$, $\frac{c\epsilon}{3\sqrt{r \log(2/\delta')}}$, $\frac{\delta}{3r}$)         ▷ ALGORITHM 4
19:      $P_{wx} \leftarrow P_{wx} + \mathcal{D}_i$.DISTANCEQUERY($0$)
20:    **end for**
21:    $\mathcal{D}_0$.INIT($\mathbf{1}_n$, $n$, $w$, $\epsilon/3$, $\delta/3$)
22:    $s_w \leftarrow s_w + \mathcal{D}_0$.DISTANCEQUERY($0$)
23: **end procedure**
24: **procedure** DISTANCEQUERY($y \in [0, R]^d$)         ▷ Lemma H.6
25:    Value $\leftarrow 0$
26:    Compute $P(y)$
27:    Compute $\|P(y)\|_2^2$
28:    **for** $i = 1 \to r$ **do**
29:      Value $\leftarrow$ Value $+ \mathcal{D}_i$.DISTANCEQUERY($P(y)_i$)        ▷ Algorithm 5
30:    **end for**
31:    Value $\leftarrow 0.5 \cdot (P_{wx} + s_w \|P(y)\|_2^2 - $ Value$)$
32:    **return** Value
33: **end procedure**
34: **end datastrucutre**

---

**Definition 4.1** (Weighted Softmax query (without normalization))**.** *For the dataset $X \in [0, R]^{n \times d}$ where $x_i^\top$ is the $i$-th row of $X$ and query $y \in [0, R]^d$, we define the weighted exponential inner product/Softmax query to be:*

$$\sum_{i \in [n]} w_i \exp(\langle x_i, y \rangle / d) = w^\top \exp(Xy/d).$$

Building on Definition 4.1, we develop a novel algorithm to answer differentially private weighted Softmax queries using the polynomial kernel method from Alman & Song (2023). Specifically, in Eq.(2), the three terms compute the weighted $\ell_2^2$ distance, which we calculate using DPTREEHIGH-DIM. By summing these terms with a controlled error, we extend DPTREEHIGHDIM to answer the Softmax query efficiently. More details can be found in Section H.

**Theorem 4.2** (Softmax query, informal version of Theorem H.7)**.** *Let $R \geq 1$. Let $r \leq \binom{2s+2d}{2s}$ and $s = O(\max\{\frac{\log(1/\epsilon_s)}{\log(\log(1/\epsilon_s)/R)}, R^2\})$. Let $\Gamma_{R,s} := \max_{j \in [s]} \frac{R^j}{\sqrt{j!}}$ (Definition H.3). Let the accuracy parameter be $\epsilon_s \in (0, 0.1)$. Our data structure DPTREESOFTMAX (Algorithm 3) uses $O(nr)$ spaces to solve Softmax query problem for dataset $X \subset [0, R]^d$ and support following operations:*

- INIT($X \subset [0, R]^d$, $n \in \mathbb{N}_+$, $w \in [-R_w, R_w]^n$, $\epsilon \in (0, 1)$, $\delta \in (0, 1)$, $\delta' \in (0, 1)$, $c \in (0, 0.1)$, $\epsilon_s \in (0, 0.1)$)*. (Algorithm 3) It takes $O(nr)$ time to initialize the data structure.*

- DISTANCEQUERY($y \in [0, R]^d$). *(Algorithm 3) It takes $O(r \log n)$ time to output a number $z$ such that*

  - *the process of output $z$ satisfies $(\epsilon, \delta + \delta')$-DP private, which computes $w^\top \exp(Xy/d)$,*
  - $|z - w^\top \exp(Xy/d)| \leq |\epsilon_s \cdot w^\top \exp(Xy/d)| + O(\epsilon^{-1} \Gamma_{R,s}^2 R_w r \sqrt{\log(1/\delta')} \cdot \log^{3/2} n)$,
  - *it holds with probability at least $0.99$.*

**Remark 4.3.** *In Theorem 4.2, the parameter $\epsilon_s$ is the accuracy parameter for polynomial kernel approximation described in Section H. Besides, note that the error bound in Theorem 4.2 does not depend on $\delta$ but depends on $\delta'$. The role of $\delta$ is to control a hidden constant term in the big $O$ notation, i.e., increasing $\delta$ reduces the error by a small constant (Fact 2.5). In practice, we set $\delta$ as a small positive constant close to $0$. Please refer to the Lemma C.7 for more details.*

### 4.4 ADAPTIVE QUERY DATA STRUCTURE

We adapt our DPTREESOFTMAX to DPTREESOFTMAXADAPTIVE (Algorithm 7) to solve the adaptive query problem. By proving it can handle any query within the query space with a certain error, we ensure it effectively processes adaptive queries. We first boost the constant probability to high probability using the Chernoff bound (Lemma B.2). Employing an $\epsilon_0$-net argument and the union bound, we bound all query points within the net. Finally, we use the Lipschitz property of the weighted Softmax distance function with an additive error to bound all points in the query space. The corresponding proofs can be found in Section G and Section H.

**Theorem 4.4** (Adaptive query Softmax data structure, informal version of Theorem H.10)**.** *Let $R \geq 1$. Let $r \leq \binom{2s+2d}{2s}$ and $s = O(\max\{\frac{\log(1/\epsilon_s)}{\log(\log(1/\epsilon_s)/R)}, R^2\})$. Let $\Gamma_{R,s} := \max_{j \in [s]} \frac{R^j}{\sqrt{j!}}$ (Definition H.3). Let the accuracy parameter be $\epsilon_s \in (0, 0.1)$. Let $X \in [0, R]^{n \times d}$ be the dataset, $w \in [-R_w, R_w]^n$ be weights, $y \in [0, R]^d$ be the query, and $p_f$ be the failure probability parameter. Let $l = O(r \log(dR/(\epsilon_s p_f)))$. There is a data structure DPTREESOFTMAXADAPTIVE (Algorithm 7) that uses $O(lnr)$ spaces to solve the weighted Softmax query problem for the dataset $X \subset [0, R]^d$ and supports the following operations:*

- INIT($X \subset [0, R]^d, n \in \mathbb{N}_+, w \in [-R_w, R_w]^n, \epsilon \in (0, 1), \delta \in (0, 1), \delta' \in (0, 1), c \in (0, 0.1), \epsilon_s \in (0, 0.1), p_f \in (0, 0.01)$). *It takes $O(lnr)$ time to initialize the data structure.*

- DISTANCEQUERY($y \in [0, R]^d$). *It takes $O(lr \log n)$ time to output a number $z$ such that*

  - *the process of output $z$ satisfies $(\epsilon, \delta + \delta')$-DP private, which computes $w^\top \exp(Xy/d)$,*
  - $|z - w^\top \exp(Xy/d)| \leq |\epsilon_s \cdot w^\top \exp(Xy/d)| + O(\epsilon^{-1} l \Gamma_{R,s}^2 R_w r \sqrt{\log(l/\delta')} \cdot \log^{3/2} n)$,
  - *it holds with probability at least $1 - p_f$ (where $p_f$ is used in $l$),*
  - *it is robust to adaptive query.*

**Remark 4.5.** *We describe the parallelization of our algorithms. In the second for loop of INIT and the for loop of DISTANCEQUERY in Algorithm 3, the $r$ DPTREEDISTANCE data structures instantiated for each coordinate are independent of each other. In addition, the for loops in Algorithm 7 are also parallelizable since the $l = O(r \log(dR/(\epsilon_s p_f)))$ copies are independent. After parallelization, we have the final time complexity of INIT to be $O(nr)$ and DISTANCEQUERY to be $O(\log n)$ in Algorithm 7 with $O(lr)$ GPU process.*

## 5 DISCUSSION

**How do we extend to self-attention and other data structures?** As self-attention is a more fundamental module in LGMs, we would like to extend our data structure to this setting. However, the challenge we faced was the dynamic update in tree nodes for each query for self-attention, which our current analysis does not support. How we can solve this challenge is crucial, and we leave it as our future direction.

Moreover, we observe that Li et al. (2015) introduces the DP matrix mechanism, which offers an alternative to our currently used binary tree data structure. A preliminary idea for extending this is as follows: consider $A = \exp(QK^\top/d)$ as defined in Definition 1.1, where $Q$ of size $m \times d$

represents the query matrix with $m$ linear queries, and $K$ serves as the database. Leveraging the results from Li et al. (2015), we could design an alternative algorithm to enhance the current binary tree data structure, DPTREE. We leave this exploration for future work.

**Why not add noise to some other places?** Where and how to add DP noises is an important problem to ask during the DP algorithm design. In this paper, we consider the problem of $\sum_{i=1}^{n} w_i \exp(\langle x_i, y \rangle / d)$ where $y, x_i \in [0, R]^d$ and $w \in [-R_w, R_w]^n$ (Definition 4.1). Notice that the only place where we add noises is in the most basic building block data structure DPTREE (Algorihtm 2). From Lemma C.3 and the way we initialize DPTREE in Algorithm 4, we see that the sensitivity $\Delta$ of this problem is $2R_w$.

A simple method for adding noise involves adding $n$ noises to a length $n$ array, with each item $w_i \exp(\langle x_i, y \rangle / d)$ for $i \in [n]$. However, this approach increases the error by a factor of $n$ by basic composition (Fact B.8) and also makes the model dependent on the number of queries. Besides, it only supports a single query and requires rebuilding the tree for each new query, rendering it impractical. In contrast, our current noise-adding technique (Lines 8 and 14 of Algorithm 2) utilizes a summation tree such that the error only increases by a factor of $\operatorname{poly} \log n$. This method also supports multiple queries, eliminating the need to rebuild the tree each time.

**How to remove the relative error parameter $\alpha$?** ~~The relative error parameter $\alpha$ in Theorem 3.1 appears because of the $(1 + \alpha)$-approximation introduced in Algorithm 4 to reduce the number of required iterations from naive $O(n)$ to $O(\log(n)/\alpha)$. However, we notice that a recent work~~ (Liu et al., 2024a) ~~does not utilize $(1 + \alpha)$-approximation and still achieves $O(\log n)$ iteration number. They introduce a new tree node representation where each node stores the sum of distances from one point to multiple points, enabling the answer to be divided into only $\log n$ values, each combining two distance values, two count values, and $y$ itself. Our DPTREE algorithms can be integrated with their method, thus removing parameter $\alpha$.~~

## 6 CONCLUSION

To our knowledge, we are the first work to provide differential privacy for cross-attention. This paper presents the DPTREE data structures, which provide a differential privacy guarantee for the cross-attention module in large generative models. This is achieved by transforming the cross-attention mechanism into a weighted distance problem. Furthermore, our algorithm is robust to adaptive queries, allowing users to interact with the model arbitrarily without extracting sensitive information from the system prompts or RAG data. Our results may inspire more privacy algorithm design in large generative models.

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

# Appendix

CONTENTS

**Roadmap.** The appendix is organized as follows. In Section A, we provide more related works. In Section B, we give the preliminary of our paper. In Section C, we give the analysis of the data structure DPTREE that can solve summation problem with DP and accuracy guarantee. In Section D, we show how to solve weighted distance problem. In Section E, we give our DPTREEDISTANCE data structure that can solve one dimensional $\ell_p^p$ distance problem with DP and accuracy guarantee. In Section F, we present the analysis of our DPTREEHIGHDIM (Algorithm 6) data structure, which can address the high-dimensional $\ell_p^p$ distance problem while ensuring differential privacy and accuracy guarantees. In Section G, we show how we can handle adaptive query. In Section H, we show how to extend our algorithm to Softmax activation and give the analysis of DPTREESOFTMAX (Algorithm 3) and DPTREESOFTMAXADAPTIVE (Algorithm 7).

## A    MORE RELATED WORK

**Differential Privacy Guarantee Analysis.** Ever since Dwork et al. (2006) proposes the notion of differential privacy (DP), it has become one of the most essential standards of privacy protection in both theoretical and empirical ways (Dwork, 2008; Li et al., 2017; Zhao & Chen, 2022; Ponomareva et al., 2023; Yang et al., 2023). DP provides a powerful, robust, and quantifiable privacy definition, allowing algorithm design with concrete privacy and accuracy guarantee (Hay et al., 2009; Esfandiari et al., 2022; Andoni et al., 2023; Li & Li, 2023b; Huang & Yi, 2021; Ghazi et al., 2023; Backurs et al., 2024; Cohen-Addad et al., 2022a; Epasto et al., 2024; Chen et al., 2022; Hopkins et al., 2023; Narayanan, 2022; 2023; Jung et al., 2019; Li & Li, 2024; Fan & Li, 2022; Fan et al., 2024; Li & Li, 2023a; Cherapanamjeri et al., 2023; Cohen-Addad et al., 2022b; Dong et al., 2024; Farhadi et al., 2022; Gopi et al., 2021; 2023; Li et al., 2022; Gopi et al., 2022; Eliáš et al., 2020; Song et al., 2023b; Dinur et al., 2023; Woodruff et al., 2023; Song et al., 2023a; Gao et al., 2024; Liang et al., 2024a; Li et al., 2024b). Additionally, new mechanisms have been proposed beyond the traditional Laplace, Gaussian, and Exponential mechanisms (Dwork & Roth, 2014). For example, truncated Laplace mechanism (Geng et al., 2020) is proved to be the current tightest the lower and upper bounds on the minimum noise amplitude and power cross all $(\epsilon, \delta)$-DP distributions.

**Cross-Attention in System Prompt, RAG, Stable Diffusion and More.** Cross-attention (Vaswani et al., 2017), first introduced in language translation, is a widely used technique in many advanced AI systems. For example, Stable Diffusion (Rombach et al., 2022; Liang et al., 2024d; Wang et al., 2023b;c; 2024b) and SORA (OpenAI, 2024b) employ cross-attention as a core module for a text-to-image conditional generation. This technique is also utilized by other multimodal models (Liang et al., 2024e), including Imagen (Saharia et al., 2022) and Diffusion Transformer (Peebles & Xie, 2023). In the realm of text-to-image editing, Hertz et al. (2022) analyzes and controls the cross-attention module to enable editing without requiring additional training. Furthermore, Yang et al. (2024) tackles the issue of inaccurate cross-attention maps, enhancing fine-grained control over edited regions while preventing unintended changes to other areas. In addition, Retrieval Augmented Generation (RAG) (Lewis et al., 2020; Borgeaud et al., 2022; Gao et al., 2023), a technique that improves model responses by retrieving information from a knowledge base or external documents, extensively uses cross-attention as its core design module. Cross-attention also has other applications. Oymak et al. (2023) demonstrates that the prompt-tuning (Liang et al., 2024c) task can be formulated as cross-attention, while Chen et al. (2021) uses cross-attention to fuse multi-scale features in vision transformers, thereby reducing computation. Moreover, attention-based Transformer architecture makes LGMs equipping many emergent ability (Wei et al., 2022), such as spatial reasoning (Wang et al., 2024a), mathematical reasoning (Li et al., 2024a), in-context learning ability (Shi et al., 2024), compositional ability (Xu et al., 2024b), few-shot adaptation ability (Shi et al., 2022b; Xu et al., 2023), and so on. There are some other works that used cross attention in Hopfield Models (Hu et al., 2023; Wu et al., 2024b; Hu et al., 2024c; Xu et al., 2024a; Wu et al., 2024a; Hu et al., 2024a;b).

## B    MORE PRELIMINARY

In Section B.1, we give the probability tools we use in the paper. In Section B.2, we provide the algebraic facts we use. In Section B.3, we give the DP facts we use in the paper. In Section B.4, we compare between popular DP mechanisms.

## B.1 PROBABILITY TOOLS

In this section, we give several probability lemmas.

**Lemma B.1** (Markov's inequality). *If $x$ is a nonnegative random variable and $t > 0$, we have*

$$\Pr[x \geq t] \leq \frac{\mathbb{E}[x]}{t}.$$

**Lemma B.2** (Chernoff bound, (Chernoff, 1952)). *Let $x_i$ be a Bernoulli random variable with probability $p_i$ of being equal to 1 and $1 - p_i$ of being equal to 0, and all $x_i$ for $i \in [n]$ are independent. Let $x = \sum_{i=1}^n x_i$. Let $\mu = \mathbb{E}[x] = \sum_{i=1}^n p_i$. Then, for all $\delta > 0$ we have*

$$\Pr[x \geq (1 + \delta)\mu] \leq \exp(-\delta^2 \mu / 3),$$

*and for all $0 < \delta < 1$*

$$\Pr[x \leq (1 - \delta)\mu] \leq \exp(-\delta^2 \mu / 2).$$

**Lemma B.3** (Chebyshev's inequality). *Let $x$ (integrable) be a random variable with finite non-zero variance $\sigma^2$ (and thus finite expected value $\mu$). Then for any real number $k > 0$,*

$$\Pr[|x - \mu| \geq k\sigma] \leq \frac{1}{k^2}.$$

## B.2 ALGEBRAIC FACTS

**Fact B.4** (Upper bound of exponential, Fact C.9 in Liang et al. (2024d)). *For $a \in \mathbb{R}$, $b \in \mathbb{R}$, $a, b \leq R$, where $R \geq 0$, we have*

$$|\exp(a) - \exp(b)| \leq \exp(R)|a - b|.$$

## B.3 DP FACTS

In this section, we present several facts about differential privacy (DP).

We first define vector neighboring dataset and sensitivity.

**Definition B.5** (Vector neighboring dataset). *We define the two neighboring datasets as $X, X' \in \mathbb{R}^n$ such that $\|X - X'\|_1 \leq 1$, i.e., they differ on a single data point.*

**Definition B.6** (Vector sensitivity). *The sensitivity of a function $f : \mathbb{R}^n \to \mathbb{R}^d$ is defined by: $\Delta := \max_{X, X' \in \mathbb{R}^n, \|X - X'\|_1 = 1} \|f(X) - f(X')\|_1$.*

We state the post-processing property, which means, in an algorithm, if one step is DP, all the following steps are DP.

**Fact B.7** (Post-processing, see Fact 2.1 in Ghazi et al. (2023)). *Let $\mathcal{A}_1$ be an $(\epsilon, \delta)$-DP algorithm and $\mathcal{A}_2$ be a (randomized) post-processing algorithm. Then the algorithm $\mathcal{A}(X) = \mathcal{A}_2(\mathcal{A}_1(X))$ is still an $(\epsilon, \delta)$-DP algorithm.*

If we have many DP algorithms, we need a composition rule. The most straightforward composition is the basic/sequential composition rule.

**Fact B.8** (Basic composition, see Fact 2.3 in Ghazi et al. (2023)). *Let $\mathcal{A}_1$ be an $(\epsilon_1, \delta_1)$-DP algorithm and $\mathcal{A}_2$ be an $(\epsilon_2, \delta_2)$-DP algorithm. Then $\mathcal{A}(X) = (\mathcal{A}_1(X), \mathcal{A}_2(\mathcal{A}_1(X), X))$ is an $(\epsilon_1 + \epsilon_2, \delta_1 + \delta_2)$-DP algorithm.*

We can do much better if we know that the inputs are disjoint.

**Fact B.9** (Parallel composition, see Fact 2.4 in Ghazi et al. (2023)). *Let $\mathcal{A}_1$ be an $(\epsilon_1, \delta_1)$-DP algorithm and $\mathcal{A}_2$ be an $(\epsilon_2, \delta_2)$-DP algorithm. Assume $\mathcal{A}_1$ and $\mathcal{A}_2$ depend on disjoint subsets of input coordinates. Then the algorithm $\mathcal{A}(X) = (\mathcal{A}_1(X), \mathcal{A}_2(\mathcal{A}_1(X), X))$ is a $(\max\{\epsilon_1, \epsilon_2\}, \max\{\delta_1, \delta_2\})$-DP algorithm.*

In addition, we have the advanced composition, which improves the dependence of the number of DP algorithms to square root but compromises the term $\delta'$.

**Theorem B.10** (Advanced composition, see Theorem 3.20 in Dwork & Roth (2014)). *For all $\epsilon, \delta, \delta' \geq 0$, the class of $(\epsilon, \delta)$-differentially private mechanisms satisfies $(\epsilon', k\delta + \delta')$-differential privacy under $k$-fold adaptive composition for:*

$$\epsilon' = k\epsilon(e^\epsilon - 1) + \epsilon\sqrt{2k \log(1/\delta')}.$$

### B.4 Comparison of Truncated Laplace, Gaussian, and Laplace Mechanisms

We first define the Laplace mechanism as below:

**Definition B.11** (Laplace distribution). *We use* $\mathrm{Lap}(b)$ *to denote the pdf:* $p(z) = \frac{1}{2b} \exp(-\frac{|z|}{b})$.

**Fact B.12.** *For* $z \sim \mathrm{Lap}(b)$, $\mathbb{E}[z] = 0$, *and* $\mathrm{Var}[z] = 2b^2$. *Furthermore, if* $b = \Delta/\epsilon$, *we have* $\mathrm{Var}[z] = 2\Delta^2/\epsilon^2$.

In this paper, we use the Chebyshev inequality to bound the error, and from Geng et al. (2020), we know that the truncated Laplace mechanism has the current minimum variance across all $(\epsilon, \delta)$-DP distributions.

The variance of Gaussian mechanism in Theorem 3.22 in Dwork & Roth (2014):

$$\mathrm{Var} = \frac{2\Delta^2 \log(1.25/\delta)}{\epsilon^2}.$$

The variance of Laplace mechanism in Fact B.12:

$$\mathrm{Var} = \frac{2\Delta^2}{\epsilon^2}.$$

The variance of truncated Laplace mechanism in Fact 2.5, for $c \in (0, 1]$:

$$\mathrm{Var} = \frac{2\Delta^2 c}{\epsilon^2}.$$

Thus, since it has the minimum variance, we choose the truncated Laplace mechanism to design our algorithms among these popular mechanisms.

## C DPTree Algorithm

In this section, we give the analysis of privacy, accuracy and runtime of our DPTree (Algorithm 2).

### C.1 Single Data Structure

We give the theorem of our DPTree data structure that can answer the summation problem with DP, accuracy, runtime guarantee.

**Theorem C.1** (DPTree data structure ). *There is a data structure (see* DPTree *in Algorithm 2) that uses* $O(n)$ *spaces to support the following operations:*

- INIT$(a \in \mathbb{R}^n, n \in \mathbb{N}_+, \Delta \in \mathbb{N}_+, \epsilon \in (0, 1), \delta \in (0, 1))$. *It takes* $O(n)$ *time to initialize the data structure.*

- QUERY$(y \in [0, R])$. *It takes* $O(\log n)$ *time to output two numbers* $z_1$ *and* $z_2$ *such that*

  - *the process satisfies* $(\epsilon, \delta)$*-DP,*
  - $|z_1 - \sum_{\{k|x_k \leq y\}} a_k| \leq O(\epsilon^{-1}\Delta \log^{3/2} n)$ *and* $|z_2 - \sum_{\{k|x_k \geq y\}} a_k| \leq O(\epsilon^{-1}\Delta \log^{3/2} n)$,
  - *it holds with probability* $0.99$.

*Proof.* The proofs follow from combining Lemma C.4 (running time of initialization), Lemma C.5 (running time of query), Lemma C.6 (DP of query), and Lemma C.7 (error of query) together. $\square$

### C.2 Boost the Constant Probability to High Probability

By applying the Chernoff bound, we can increase the probability of obtaining a correct result. This is achieved by replicating the data structure multiple times, generating several independent results, and then reporting the median of these results. Taking the median helps mitigate the effect of outliers and ensures that the final answer is reliable with high probability.

**Theorem C.2** (High-probability)**.** *There is a data structure that uses $O(n \log(1/\delta_{\mathrm{fail}}))$ spaces to support the following operations*

- INIT($a \in \mathbb{R}^n, n \in \mathbb{N}_+, \Delta \in \mathbb{N}_+, \epsilon \in (0,1), \delta \in (0,1), \delta_{\mathrm{fail}} \in (0, 0.01)$). *It takes $O(n \log(1/\delta_{\mathrm{fail}}))$ time to initialize the data structure.*

- QUERY($y \in [0, R]$). *It takes $O(\log(n) \cdot \log(1/\delta_{\mathrm{fail}}))$ time to two numbers $z_1$ and $z_2$ such that*

  - *the process satisfies $(\epsilon, \delta)$-DP,*
  - $|z_1 - \sum_{\{k | x_k \leq y\}} a_k| \leq O(\epsilon^{-1} \Delta \log^{3/2}(n) \cdot \log(1/\delta_{\mathrm{fail}}))$ *and* $|z_2 - \sum_{\{k | x_k \geq y\}} a_k| \leq O(\epsilon^{-1} \Delta \log^{3/2}(n) \cdot \log(1/\delta_{\mathrm{fail}}))$,
  - *it holds with probability $1 - \delta_{\mathrm{fail}}$ for failure probability $\delta_{\mathrm{fail}} \in (0, 0.01)$.*

*Proof.* Note that our data structure (Theorem C.1) succeeds with probability 0.99. The success of the algorithm (Theorem C.1) can be viewed as a Bernoulli random variable, to which we apply the Chernoff bound (Lemma B.2). By repeating the data structure $O(\log(1/\delta_{\mathrm{fail}}))$ times and taking the median of the outputs, we boost the success probability. The details are following.

To boost the success probability, we assume the query is repeated $l$ times. Let $i \in [l]$, and let $z_i$ denote the indicator random variable for the success of the $i$-th instance of the data structure for a single query. Let $z = \sum_{i=1}^{l} z_i$ be the total success times. Since $p = \Pr[z_i = 1] = 0.99$, we can have $\mu = \mathbb{E}[z] = \sum_{i=1}^{l} p = lp$. Note that $p = 0.99$. By setting $\delta = 0.1$ and using Chernoff bound from Lemma B.2, we can show

$$\Pr[z \leq l/2] \leq \Pr[z \leq (1-\delta)lp] \leq \exp(-\delta^2 lp/2).$$

Note that we want $z > l/2$ (since we want at least half to succeed so we could take the median),

$$\Pr[z > l/2] \geq 1 - \exp(-\delta^2 lp/2).$$

To ensure that failure probability is $\delta_{\mathrm{fail}}$, we have

$$\exp(-\delta^2 lp/2) = \delta_{\mathrm{fail}}.$$

We can make this hold by choosing $l = O(\log(1/\delta_{\mathrm{fail}}))$.

By the DP basic composition rule (Fact B.8), we need to choose $\epsilon = \epsilon'/O(\log(1/\delta_{\mathrm{fail}}))$ and $\delta = \delta'/O(\log(1/\delta_{\mathrm{fail}}))$ where $\epsilon', \delta'$ are the $\epsilon, \delta$ in Theorem C.1. □

## C.3 SENSITIVITY FOR SUMMATION PROBLEM

Our DP summation tree data structure DPTREE (Algorithm 2) requires sensitivity parameter $\Delta$. In this section, we show that for the summation problem, we have the sensitivity $\Delta = 2R$ if the input $X \in [-R, R]^n$.

**Lemma C.3** (Sensitivity of summation)**.** *Let $X \in [-R, R]^n$. We have the sensitivity $\Delta = 2R$ for DPTREE.INIT in Algorithm 2.*

*Proof.* Let's say two neighboring datasets $X$ and $X'$ differ in $x_i$ and $x_i'$ for some $i$ in the array $X$. Then for a summation problem, i.e. $f(X) := \sum_{i=1}^{n} x_i$, we have

$$\Delta = \max_{X, X'} |f(X) - f(X')| = \max_{X, X'} |x_i - x_i'| = 2R.$$

where the first step follows from Definition B.6, the second step follows from $X, X'$ differ in $x_i, x_i'$, and the last step follows from each coordinate of the dataset is bounded in $[-R, R]$. □

## C.4 ALGORITHM OF DATA STRUCTURE

In this section, we analyze the accuracy, DP, and runtime of Algorithm 2.

We first analyze the runtime.

**Lemma C.4** (Runtime of initialization, Algorithm 2). *For the initialization, we have the time complexity of Algorithm 2 is $O(n)$.*

*Proof.* All the computations are dominated by $O(n)$ time. □

**Lemma C.5** (Runtime of query, Algorithm 2). *For each query, we have the time complexity of Algorithm 2 is $O(\log n)$.*

*Proof.* Due to the property of tree, we will use at most $2 \log n$ nodes in the tree, thus the running time is $O(\log n)$. □

We now analyze the DP.

**Lemma C.6** (Privacy of query, Algorithm 2). *The output process of* QUERY *(see Algorithm 2) is $(\epsilon, \delta)$-DP.*

*Proof.* Suppose that our dataset is $X \in [-R, R]^n$. Note that we only add noise in the pre-processing stage. There is no noise in the query stage. Since the problem we care about is summation, if we change one leaf node, the sensitivity $\Delta = 2R$ (see Lemma C.3). Since we add noise to each node in the tree, and each leaf node count will contribute to $\log n$ nodes, it is equivalent to our output function being in $\log n$ dimension. We will then blow up the DP parameter by $\log n$ factor. Thus, using the basic composition rule (Fact B.8), the DP guarantee for the whole tree data structure is $((\epsilon / \log n) \cdot \log n, (\delta / \log n) \cdot \log n)$ which is $(\epsilon, \delta)$-DP. □

We now analyze the accuracy.

**Lemma C.7** (Accuracy of query, Algorithm 2). *Let $\epsilon \in (0, 1)$ and $\delta \in (0, 1)$. Then, using Chebyshev's inequality and Fact 2.5, we have the error of* QUERY*(see Algorithm 2) output is upper bounded by:*

$$O(\epsilon^{-1} \Delta \log^{3/2} n).$$

*with probability* 0.99.

*Proof.* Let $y \in [0, R]$ be the query. Let $A_1, A_2 = $ QUERY$(y)$ denote the noised query answers returned by DPTREE.QUERY in Algorithm 2. Let $A_1^*, A_2^*$ be the true query answers without noise. Let $z := A_1 - A_1^* + A_2 - A_2^*$, which from Algorithm 2 we can see this is the sum of $O(\log n)$ independent truncated Laplace random variables each with parameter $\text{TLap}(\Delta, \epsilon / \log n, \delta / \log n)$. Thus,

$$z = \sum_{i=1}^{O(\log n)} z_i$$

where $z_i \sim \text{TLap}(\Delta, \epsilon / \log n, \delta / \log n)$, and every $z_i$ are independent to each other.

We know $\mu = \mathbb{E}[z] = 0$ since $\mathbb{E}[z_i] = 0$. From Fact 2.5, we know the variance for each $z_i$ is $\text{Var}[z_i] = c\epsilon^{-2}\Delta^2 \log^2 n$ where $0 < c \leq 2$ and $c = 2$ when $\delta = 0$.

Therefore, we can show

$$\text{Var}[z] = \text{Var}\left[\sum_{i=1}^{O(\log n)} z_i\right]$$
$$= \sum_{i=1}^{O(\log n)} \text{Var}[z_i]$$
$$= O(c\epsilon^{-2}\Delta^2 \log^3 n) \tag{4}$$

where the first step follows from definition of $z$, the second step follows from every $z_i$ are independent to each other, and the last step follows from $\text{Var}[z_i] = O(c\epsilon^{-2}\Delta^2 \log^2 n)$.

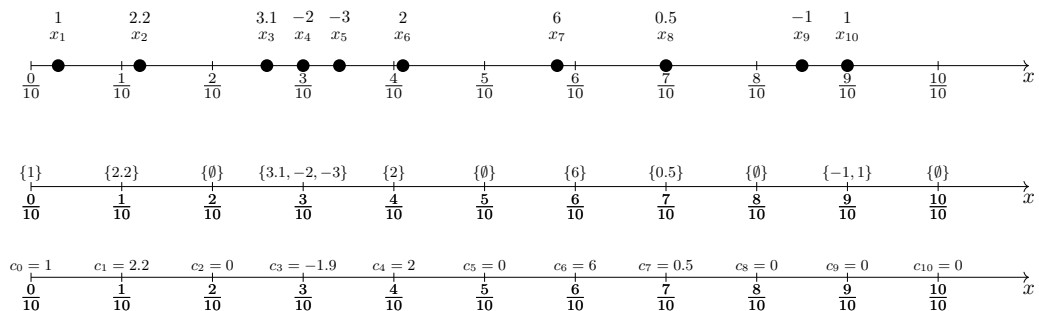

Figure 1: The visualization of how to compute the weighted $\ell_1$ distance for rounded dataset $X \in [0,1]^{10}$. The number above each $x_i$ is $w_i$. See Algorithm 4 for details. Suppose $y = 0$. Then $\sum_{i=1}^{n} w_i |y - x_i| = 0.1 \cdot 2.2 + 0.3 \cdot 3.1 + 0.3 \cdot (-2) + 0.3 \cdot (-3) + 0.4 \cdot 2 + 0.6 \cdot 6 + 0.7 \cdot 0.5 + 0.9 \cdot (-1) + 0.9 \cdot 1 = 4.4$. See more details in Lemma D.1.

Note that we wish to bound $|z|$ as our error.

Using Lemma B.3, we can have

$$\Pr[|z| \geq k\sigma] \leq \frac{1}{k^2}.$$

We know that $\sigma = \sqrt{\mathrm{Var}[z]} = O(c^{1/2}\epsilon^{-1}\Delta \log^{3/2} n)$. Picking $k = 10$, we have

$$\Pr[|z| < 10\sigma] \geq 0.99.$$

Thus, we conclude that error is bounded by $O(c^{1/2}\epsilon^{-1}\Delta \log^{3/2} n) = O(\epsilon^{-1}\Delta \log^{3/2} n)$ (since $c \in (0,2]$) with probability 0.99. $\qquad\square$

## D   WEIGHTED $\ell_p^p$ DISTANCE

In this section, we introduce how to handle weighted $\ell_p^p$ distance problem in the high level idea. We can solve high dimensional weighted problem by decomposing each coordinate of the high dimensional dataset. Thus, we only need to show how to solve the one-dimensional weighted problem.

For data in $d$-dimension, due to the decomposability of $\ell_p^p$ distance, our problem will be: given $x_i \in [0, R]^d$ and $w_i \in \mathbb{R}$ for $i \in [n]$, and $y \in [0, R]^d$, we can compute

$$\sum_{i=1}^{n} w_i \cdot \|y - x_i\|_p^p = \sum_{j=1}^{d} \sum_{i=1}^{n} w_i \cdot |y_j - x_{i,j}|^p$$

where $x_{i,j}, y_j$ means the $j$-th coordinates of $x_i, y$ for $j \in [d]$.

Now we can give the lemma for weighted distance of dataset.

**Lemma D.1** (Weighted distance one dimension). *For a collection of numbers $\{x_1, x_2, \cdots, x_n\} \subset \mathbb{R}$ and corresponding weights $\{w_1, w_2, \cdots, w_n\} \subset \mathbb{R}$, and a number $y \in \mathbb{R}$. We define two sets*

$$S_+ := \{k \in [n] \ : \ x_k > y\}$$
$$S_- := \{k \in [n] \ : \ x_k < y\},$$

*It holds*

$$\sum_{k=1}^{n} w_k |x_k - y|^p = \sum_{j=0}^{p} \binom{p}{j} y^{p-j}((-1)^{p-j} \sum_{k \in S_+} w_k x_k^j + (-1)^j \sum_{k \in S_-} w_k x_k^j),$$

*where $\binom{p}{j}$ denotes the binomial coefficient that $\binom{p}{j} = \frac{p!}{j!(p-j)!}$.*

*Proof.* We show that

$$\sum_{k=1}^{n} w_k |x_k - y|^p = \sum_{x_k \in S_+} w_k (x_k - y)^p + \sum_{x_k \in S_-} w_k (y - x_k)^p$$

$$= (\sum_{x_k \in S_+} w_k \sum_{j=0}^{p} (-1)^{p-j} \binom{p}{j} x_k^j y^{p-j}) + (\sum_{x_k \in S_-} w_k \sum_{j=0}^{p} (-1)^j \binom{p}{j} x_k^j y^{p-j})$$

$$= \sum_{j=0}^{p} (\binom{p}{j}(-1)^{p-j} y^{p-j} \sum_{k \in S_+} w_k x_k^j) + \sum_{j=0}^{p} (\binom{p}{j}(-1)^j y^{p-j} \sum_{k \in S_-} w_k x_k^j)$$

$$= \sum_{j=0}^{p} \binom{p}{j} y^{p-j} ((-1)^{p-j} \sum_{k \in S_+} w_k x_k^j + (-1)^j \sum_{k \in S_-} w_k x_k^j).$$

Thus, we complete the proof. $\square$

## E  ONE-DIMENSIONAL WEIGHTED $\ell_p^p$ DISTANCE QUERY

In this section, we generalize the algorithms in Backurs et al. (2024) and Liu et al. (2024a) to weighted distance. Here, we compute the problem of one-dimensional weighted $\ell_p^p$ distance query i.e. $\sum_{i \in [n]} w_i |y - x_i|$ for a given query $y \in [0, R]$, weights $w \in [-R_w, R_w]^n$ and dataset $X \subset [0, R]$ and $n = |X|$. In this section, we give the theorem for our DPTREEDISTANCE data structure.

---

**Algorithm 4** Pre-processing data structure

1: **datastructure** DPTREEDISTANCE      ▷ Theorem E.1
2: **members**
3:    $\mathcal{D}_0, \ldots, \mathcal{D}_p :$ DPTREE      ▷ Alg. 2
4:    $X : [0, R]^n$
5:    $w : [-R_w, R_w]^n$
6: **end members**
7: **procedure** INIT($X \subset [0, R], n \in \mathbb{N}_+, w \in [-R_w, R_w]^n, \epsilon \in (0,1), \delta \in (0,1)$ ) ▷ Lemma D.1
8:    $X, w, a \leftarrow X, w, 0^{n \times (p+1)}$
9:    **for** $i = 1 \to n$ **do**      ▷ $x_i \in X$ for $i \in [n]$
10:      Let $j \in [n]$ denotes the integer such that $x_i \in [(j-1)R/n, jR/n)$
11:      **for** $q = 0 \to p$ **do**
12:        $a_{j,q} \leftarrow a_{j,q} + w_i x_i^q$
13:      **end for**
14:    **end for**
15:    **for** $q = 0 \to p$ **do**
16:      $\mathcal{D}_q.\text{INIT}(a_{:,q}, n, 2R_w R^q, \epsilon/(p+1), \delta/(p+1))$      ▷ Alg. 2, Lemma C.3
17:    **end for**
18: **end procedure**
19: **end datastructure**

---

**Algorithm 5** One dimensional weighted $\ell_p^p$ distance query

1: **datastructure** DPTREEDISTANCE      ▷ Theorem E.1
2: **procedure** DISTANCEQUERY($y \in [0, R]$)
3:    **for** $q = 0 \to p$ **do**
4:      $c_{\text{left},q}, c_{\text{right},q} \leftarrow \mathcal{D}_q.\text{QUERY}(y)$
5:    **end for**
6:    **return** $\sum_{q=0}^{p} \binom{p}{q} y^{p-q} ((-1)^{p-q} c_{\text{right},q} + (-1)^q c_{\text{left},q})$
7: **end procedure**
8: **end datastructure**

**Theorem E.1** (DPTREEDISTANCE data structure ). *There is a data structure* DPTREEDISTANCE *(Algorithm 4,5) that uses $O(np)$ spaces to solve weighted $\ell_p^p$ distance query problem for dataset $X \subset [0, R]$ and support the following operations:*

- INIT($X \subset [0, R], n \in \mathbb{N}_+, w \in [-R_w, R_w]^n, \epsilon \in (0, 1), \delta \in (0, 1)$). *(Algorithm 4) It takes $O(np)$ time to initialize the data structure.*

- DISTANCEQUERY($y \in [0, R]$). *(Algorithm 5) It takes $O(p \log n)$ time to output a number $z$ such that*

    – *the process of output $z$ satisfies $(\epsilon, \delta)$-DP private, which computes $\sum_{i \in [n]} w_i |y - x_i|$,*

    – $|z - \sum_{i \in [n]} w_i |y - x_i|| \leq O(\epsilon^{-1} p R_w (2R)^p \log^{3/2} n)$,

    – *it holds with probability $0.99$.*

*Proof.* We set the total layers of one tree $L = (\log n)$. There are $p + 1$ trees.

**Init Time and Space.** The total number of nodes on one tree is $O(n)$. There are total $O(pn)$ values stored for $p + 1$ trees. Adding the time of iterating all data points, initializing these values takes $O(pn)$ time.

**Query Time.** Each query iterates through all layers. On each layer it takes $O(1)$ time to calculate $c_{\text{left},q}$ and $c_{\text{right},q}$. There are $(\log n)$ layers, and $p + 1$ trees, so the total query time is $O(p \log n)$.

**Privacy Guarantees.** For each $\mathcal{D}_q$ for $q \in \{0, 1, \ldots, p\}$, we input $a_{:,q}$. Since $X \in [0, R]^n$ and $w \in [-R_w, R_w]^n$, the input range for $a_{:,q}$ is $[-R_w R^q, R_w R^q]$. Then from Lemma C.3, sensitivity is $2R_w R^q$.

From Lemma C.6, we know each $\mathcal{D}_q$ query is $(\epsilon/(p + 1), \delta/(p + 1))$-DP. By basic composition Fact B.8, the total differential privacy parameter is $(\epsilon, \delta)$. This completes the proof.

**Error Guarantees.** The additive error consists of two parts.

The first part is from the data in the leaf node which contains query $y$. The error is

$$\sum_{x_k \in [(j-1) \cdot R/2^L, j \cdot R/2^L)} |x_k - y|^p \leq n \cdot (\frac{R}{2^L})^p.$$

When $L = \log n$, this error is $O(R^p / n^{p-1})$.

The second part is the Truncated Laplace noise. From the proof of Lemma C.7, we have each $\mathcal{D}_q$ for $q \in \{0, 1, \ldots, p\}$ has $O(L)$ independent $\text{TLap}(\Delta_q, \epsilon_q/L, \delta_q/L)$ noises for $L = \log n$ layers.

Let $A$ be the noisy output of DISTANCEQUERY in Algorithm 5 and $A_* = \sum_{k \in [n]} w_k |y - x_k|$ be the true output. Then, for our Algorithm 4 and 5, the variance is

$$\text{Var}[\sum_i^L \text{TLap}(\Delta_q, \epsilon_q/L, \delta_q/L)] = \sum_i^L \text{Var}[\text{TLap}(\Delta_q, \epsilon_q/L, \delta_q/L)]$$
$$= O(L^3 \epsilon_q^{-2} \Delta_q^2)$$

Replacing $\Delta_q = O(R^q R_w)$ and $\epsilon_q = O(\epsilon/p)$, using Lemma B.3, with high probability $0.99$, we have

$$|\sum_i^L \text{TLap}(\Delta_q, \epsilon_q/L, \delta_q/L)| \leq O(p R_w R^q L^{3/2}/\epsilon). \tag{5}$$

Then we bound the error with this inequality:

$$|A - A'| \leq |\sum_{q=0}^p \binom{p}{q} y^{p-q} \sum_{i=1}^L ((-1)^{p-q} \text{TLap}(\Delta_q, \epsilon_q/L, \delta_q/L) + (-1)^q \text{TLap}(\Delta_q, \epsilon_q/L, \delta_q/L))|$$

$$\leq \sum_{q=0}^{p} \binom{p}{q} y^{p-q} | \sum_{i=1}^{L} (\text{TLap}(\Delta_q, \epsilon_q/L, \delta_q/L) + \text{TLap}(\Delta_q, \epsilon_q/L, \delta_q/L))|$$

$$= \sum_{q=0}^{p} \binom{p}{q} y^{p-q} \cdot O(pR_w R^q L^{3/2}/\epsilon)$$

$$= O(\epsilon^{-1} pR_w L^{3/2} \sum_{q=0}^{p} \binom{p}{q} y^{p-q} R^q)$$

$$= O(\epsilon^{-1} pR_w L^{3/2} (y+R)^p)$$

$$= O(\epsilon^{-1} pR_w (2R)^p \log^{3/2} n),$$

where the third step follows from Eq. (5), and the last step is from $L = \log n$ and $y \in [0, R]$.

Therefore, by triangle inequality and two parts of error, the total error is

$$O(R^p/n^{p-1}) + O(\epsilon^{-1} pR_w (2R)^p \log^{3/2} n) \leq O(\epsilon^{-1} pR_w (2R)^p \log^{3/2} n),$$

since $p \geq 1$ and $n \in \mathbb{N}_+$. This completes the proof. □

## F  HIGH-DIMENSIONAL WEIGHTED $\ell_p^p$ QUERY

In this section, we show how we can solve the high dimensional weighted $\ell_p^p$ distance problem, generalizing results from Backurs et al. (2024) and Liu et al. (2024a). In Section F.1, we give the analysis of Algorithm 6. In Section F.2, we give the theorem of our DPTREEHIGHDIM data structure.

Algorithm 4,5 can be naturally extended to higher dimensions because of the decomposability of the $\ell_p^p$ distance function. We construct $d$ separate one-dimensional distance query data structures, each corresponding to a coordinate projection of the dataset.

### F.1  PRIVACY AND ACCURACY ANALYSIS FOR HIGH DIMENSIONAL WEIGHTED DISTANCE

We now give the analysis of our Algorithm 6 for high dimensional weighted $\ell_p^p$ distance query.

---
**Algorithm 6** High-dimensional weighted $\ell_p^p$ distance query
---
1: **datastrucutre** DPTREEHIGHDIM                                  ▷ Theorem F.3
2: **members**
3:     $\mathcal{D}_1, \ldots, \mathcal{D}_d$ : DPTREEDISTANCE          ▷ Alg. 4
4:     $X : [0, R]^{n \times d}$
5:     $w : [-R_w, R_w]^n$
6: **end members**
7: **procedure** INIT($X \subset [0, R]^d$, $n \in \mathbb{N}_+$, $w \in [-R_w, R_w]^n$, $\epsilon \in (0,1)$, $\delta \in (0,1)$, $\delta' \in (0,1)$, $c \in (0, 0.1)$)
8:     $X \leftarrow X$
9:     $w \leftarrow w$
10:     **for** $i = 1 \to d$ **do**
11:         $\mathcal{D}_i$.INIT($X_{:,i}$, $n$, $w$, $c\epsilon/\sqrt{d \log(1/\delta')}$, $\delta/d$)     ▷ Alg. 4
12:     **end for**
13: **end procedure**
14: **procedure** DISTANCEQUERY($y \in [0, R]^d$)        ▷ Lemma F.1, Lemma F.2
15:     Value $\leftarrow 0$
16:     **for** $i = 1 \to d$ **do**
17:         Value $\leftarrow$ Value $+ \mathcal{D}_i$.DISTANCEQUERY($y_i$)     ▷ Alg. 5
18:     **end for**
19:     **return** Value
20: **end procedure**
21: **end datastrucutre**
---

**Lemma F.1** (Privacy of DISTANCEQUERY, Algorithm 6). *If the following conditions hold*

- *Let data set $X \in [0, R]^{n \times d}$, weights $w \in [-R_w, R_w]^n$, query $y \in [0, R]^d$.*

- *Let $\epsilon \in (0, 1)$, $\delta \in (0, 1)$, $\delta' \in (0, 1)$.*

- *Let $c \in (0, 0.1)$ be a small constant and $A$ be the output of DISTANCEQUERY in Algorithm 6, where each one-dimensional algorithm is configured to be $(c\epsilon/\sqrt{d\log(1/\delta')}, \delta/d)$-DP (see Line 11).*

- *Let $A_* = \sum_{i \in [n]} w_i \|y - x_i\|_p^p$ represent the true distance query value.*

- *Let $\epsilon = O(\log(1/\delta'))$.*

*Then, we have the output process of DISTANCEQUERY (Algorithm 6) is $(\epsilon, \delta + \delta')$-DP.*

*Proof.* The $(\epsilon, \delta + \delta')$-DP guarantee follows from the approximate DP advanced composition result Theorem B.10. Our algorithm instantiate each one-dimensional data structure with $(c\epsilon/\sqrt{d\log(1/\delta')}, \delta/d)$-DP total $d$ times.

From advanced composition in Theorem B.10, for a sufficient small parameter $\epsilon$ and constant $c$, we have the final privacy loss parameter be:

$$O(c\epsilon\sqrt{2d\log(1/\delta')}/\sqrt{d\log(1/\delta')}) = O(\epsilon)$$

and the final failure probability parameter be:

$$d\delta/d + \delta' = \delta + \delta'.$$

$\square$

**Lemma F.2** (Accuracy of DISTANCEQUERY, Algorithm 6). *If the following conditions hold*

- *Let data set $X \in [0, R]^{n \times d}$, weights $w \in [-R_w, R_w]^n$, query $y \in [0, R]^d$.*

- *Let $\epsilon \in (0, 1)$, $\delta \in (0, 1)$, $\delta' \in (0, 1)$.*

- *Let $c \in (0, 0.1)$ be a small constant and $A$ be the output of DISTANCEQUERY in Algorithm 6, where each one-dimensional algorithm is configured to be $(c\epsilon/\sqrt{d\log(1/\delta')}, \delta/d)$-DP (see Line 11).*

- *Let $A_* = \sum_{i \in [n]} w_i \|y - x_i\|_p^p$ represent the true distance query value.*

*With probability* 0.99, *we have*

$$|A - A_*| \leq O(\epsilon^{-1} dp(2R)^p R_w \sqrt{\log(1/\delta')} \cdot \log^{3/2} n).$$

*Proof.* Let $A_i$ be the $i$-th dimension output returned by $\mathcal{D}_i$ in Algorithm 6. Let $A_{*,i}$ be the true distance query value in the $i$-th dimension. Observe that $A_* = \sum_{i=1}^d A_{*,i}$ and $A = \sum_{i=1}^d A_i$.

We follow the similar idea in the proof of Theorem E.1. With $\epsilon$ scaled down by $c\epsilon/\sqrt{d\log(1/\delta')}$ and $\delta$ scaled down by $\delta/d$, the variance of each individual dimension is given by (see proof of Theorem E.1)

$$O(\epsilon^{-2} dp^2 (2R)^{2p} R_w^2 \log(1/\delta') \log^3 n).$$

Thus, the total variance for $d$ instantiated data structures is then

$$O(\epsilon^{-2} d^2 p^2 (2R)^{2p} R_w^2 \log(1/\delta') \log^3 n).$$

Finally, from Lemma B.3, we have the additive error given by

$$O(\epsilon^{-1} dp(2R)^p R_w \sqrt{\log(1/\delta')} \cdot \log^{3/2} n).$$

$\square$

## F.2 HIGH DIMENSION SINGLE DATA STRUCTURE

We have the data structure that can solve weighted $\ell_p^p$ distance problem in $d$-dimensional data.

**Theorem F.3** (DPTREEHIGHDIM data structure). *There is a data structure* DPTREEHIGHDIM *(Algorithm 6) that uses $O(npd)$ spaces to solve weighted $\ell_p^p$ distance query problem for dataset $X \subset [0, R]^d$ and support the following operations:*

- INIT($X \subset [0, R]^d, n \in \mathbb{N}_+, w \in [-R_w, R_w]^n, \epsilon \in (0,1), \delta \in (0,1), \delta' \in (0,1), c \in (0, 0.1)$). *(Algorithm 6) It takes $O(npd)$ time to initialize the data structure.*

- DISTANCEQUERY($y \in [0, R]^d$). *(Algorithm 6) It takes $O(dp \log n)$ time to output a number $z$ such that*

  - *the process of output $z$ satisfies is $(\epsilon, \delta + \delta')$-DP private, which computes $\sum_{i \in [n]} w_i \|y - x_i\|_p^p$,*
  - $|z - \sum_{i \in [n]} w_i \|y - x_i\|_1| \le O(\epsilon^{-1} dp(2R)^p R_w \sqrt{\log(1/\delta')} \cdot \log^{3/2} n)$,
  - *it holds with probability 0.99.*

*Proof.* For the runtime analysis, since we loop data structure DPTREEDISTANCE $d$ times, an additional $d$ factor will appear for both initialization and query time complexity. The DP is proved by Lemma F.1. The accuracy is proved by Lemma F.2. $\qquad\square$

# G ADAPTIVE QUERY

In this section, we introduce how we can solve the adaptive query problem by our algorithm, using some tools from Qin et al. (2022). Our idea is that, if we can prove that our algorithm can solve any query in the query space with certain error. Then, since adaptive query must lie in this space, we can handle adaptive query. In Section G.1, we show how we can boost the constant probability of our algorithm to high probability. In Section G.2, we show how we can apply the notion of $\epsilon_0$-net and bound all query points in net. In Section G.3, we show how we can bound all points in the query space by introducing an additive error.

First, from Theorem F.3, given query $y \in [0, R]^d$ we have DISTANCEQUERY($y$) that can solve $d$-dimension weighted $\ell_p^p$ distance problem with constant probability 0.99. Now we show how to improve it to solve adaptive query problem. Here, we focus on the case when $p = 1$.

## G.1 BOOST THE CONSTANT PROBABILITY TO HIGH PROBABILITY

We can repeat the data structure multiple times and take the median to boost the constant probability using Chernoff bound from Lemma B.2.

**Lemma G.1** (Using Chernoff bound to boost the probability). *If the following conditions hold:*

- *Let data set $X \in [0, R]^{n \times d}$, weights $w \in [-R_w, R_w]^n$, query $y \in [0, R]^d$.*

- *Let the failure probability $p_f \in (0, 0.01)$.*

- *We create $l = O(\log(1/p_f))$ independent copies of data structure* DPTREEHIGHDIM *and take the median of the outputs with each data structure instantiated with $(\epsilon/l, (\delta + \delta')/l)$-DP.*

- *Let $B = O(\epsilon^{-1} l R R_w d \sqrt{\log(l/\delta')} \cdot \log^{3/2} n)$.*

*Then for each fixed query point $y$, we can have the process of outputting the median of $l$ responses is $(\epsilon, \delta + \delta')$-DP and the error is upper bounded by $B$ with probability $1 - p_f$.*

*Proof.* By basic composition Fact B.8, we prove the DP. Similar to the proof of Theorem C.2, we prove the error by Chernoff bound (Lemma B.2). $\qquad\square$

### G.2 FROM EACH FIXED QUERY POINT TO ALL ON-NET POINTS

In this section, we build $\epsilon_0$-net and generalize from each fixed query point to all on-net points.

**Definition G.2** ($\ell_p$ $\epsilon_0$-net, see Definition 4.2.1 in Vershynin (2017)). *We define $N$ be $\ell_p$ $\epsilon_0$-net of $\mathcal{B} := \{q \in [0, R]^d\}$ such that, for every point $q$ in $\mathcal{B}$, there exists $y \in N$ satisfying $\|y - q\|_p \leq \epsilon_0$.*

**Fact G.3** ($\ell_\infty$ $\epsilon_0$-net). *Let $N$ be the $\ell_\infty$ $\epsilon_0$-net of $\mathcal{B}$, and $|N|$ be the size of net $N$. We have $|N| \leq (5R/\epsilon_0)^d$.*

**Fact G.4** ($\ell_2$ $\epsilon_0$-net, see Lemma 5 in Woodruff (2014)). *Let $N$ be the $\ell_2$ $\epsilon_0$-net of $\mathcal{B}$, and $|N|$ be the size of net $N$. We have $|N| \leq (5R/\epsilon_0)^d$.*

**Fact G.5** ($\ell_1$ $\epsilon_0$-net, see Theorem 2 in Guntuboyina & Sen (2012)). *Let $N$ be the $\ell_1$ $\epsilon_0$-net of $\mathcal{B}$, and $|N|$ be the size of net $N$. We have $|N| \leq (5R\sqrt{d}/\epsilon_0)^d$.*

**Lemma G.6** (From for each query point to for all points in net). *If the following conditions hold:*

- *Let $N$ be the $\ell_\infty$ $\epsilon_0$-net of $\mathcal{B}$, and $|N|$ be the size of net $N$.*

- *Let data set $X \in [0, R]^{n \times d}$, weights $w \in [-R_w, R_w]^n$, query $y \in [0, R]^d$.*

- *Let the failure probability $p_f \in (0, 0.01)$.*

- *We create $l = O(\log(|N|/p_f))$ independent copies of data structure DPTREEHIGHDIM and take the median of the outputs with each data structure instantiated with $(\epsilon/l, (\delta + \delta')/l)$-DP.*

- *Let $B = O(\epsilon^{-1}lRR_wd\sqrt{\log(l/\delta')} \cdot \log^{3/2} n)$.*

*Then with probability $1 - p_f$, for all query points $y \in N$, we can have the process of outputting the median of $l$ responses is $(\epsilon, \delta + \delta')$-DP and the error is upper bounded by $B$.*

*Proof.* By basic composition Fact B.8, we prove the DP. From Lemma G.1, we know for each $y \in N$, the error is upper bounded by $B$ with probability $1 - p_f/|N|$.

Then, by union bound, with probability $1 - p_f$, the error of all $|N|$ query points in the net $y \in N$ is upper bounded by $B$. $\qquad\square$

### G.3 FROM NET POINTS TO ALL POINTS

In this section, we show how to generalize points from net to all points in the query space. Since adaptive query must lie in this space, we complete the proof of adaptive query.

**Lemma G.7** (Lipschitz of query function). *If the following conditions hold:*

- *Let data set $X \in [0, R]^{n \times d}$, weights $w \in [-R_w, R_w]^n$, query $y \in [0, R]^d$.*

- *Let $Z(y) := \sum_{i \in [n]} w_i\|y - x_i\|_1$.*

- *Let $L = nR_w$.*

*Then, we have $Z(y)$ is $L$-Lipschitz (note that we have $\ell_1$ Lipschitz here).*

*Proof.* We can show
$$|Z(y) - Z(\widetilde{y})| = |\sum_{i \in [n]} w_i\|y - x_i\|_1 - \sum_{i \in [n]} w_i\|\widetilde{y} - x_i\|_1|$$
$$\leq \sum_{i \in [n]} |w_i| \cdot |\|y - x_i\|_1 - \|\widetilde{y} - x_i\|_1|$$
$$\leq \sum_{i \in [n]} |w_i| \cdot \|y - \widetilde{y}\|_1$$
$$= nR_w \cdot \|y - \widetilde{y}\|_1$$

where the first step follows from definition of $Z(y)$, the second step follows from triangular inequality, the third step follows from reverse triangular inequality, the fourth step follows from $w \in [-R_w, R_w]^n$. $\qquad\square$

**Lemma G.8** (From points in net to all points in query space). *If the following conditions hold:*

- *Let $N$ be the $\ell_\infty$ $\epsilon_0$-net of $\mathcal{B}$, and $|N|$ be the size of net $N$.*

- *Let data set $X \in [0, R]^{n \times d}$, weights $w \in [-R_w, R_w]^n$, query $y \in [0, R]^d$.*

- *Let the failure probability $p_f \in (0, 0.01)$.*

- *We create $l = O(\log((R/\epsilon_0)^d/p_f))$ independent copies of data structure $\{\text{DPTREEHIGHDIM}_j\}_{j=1}^l$ and take the median of the outputs with each data structure instantiated with $(\epsilon/l, (\delta + \delta')/l)$-DP.*

- *Let $f(y) := \text{Median}(\{\text{DPTREEHIGHDIM}_j.\text{DISTANCEQUERY}(y)\}_{j=1}^l)$.*

- *Let $Z(y) := \sum_{i \in [n]} w_i \|y - x_i\|_1$, where $Z(y)$ is L-Lipschitz with $L = nR_w$.*

- *Let $B = O(\epsilon^{-1} l R R_w d \sqrt{\log(l/\delta')} \cdot \log^{3/2} n)$.*

*Then with probability $1 - p_f$, for all query points $q \in \mathcal{B}$, there exists a point $y \in N$ which is the closest to $q$, we can have the process of outputting the median of $l$ responses is $(\epsilon, \delta + \delta')$-DP and the error satisfy*

$$|f(y) - Z(q)| \le B + Ld\epsilon_0.$$

*Proof.* By basic composition Fact B.8, we prove the DP.

We define an event $E$ such that:

$$\forall y \in N$$
$$|f(y) - Z(y)| \le B.$$

From Lemma G.1, with $l = O(\log(|N|/p_f))$ we know

$$\Pr[\text{event } E \text{ holds}] \ge 1 - p_f$$

We can show

$$l = O(\log(|N|/p_f)$$
$$= O(\log((R/\epsilon_0)^d/p_f)$$

where the first step follows from definition of $l$, the second step follows from Fact G.3.

We condition on event $E$ to be held. Then, by definition of $\ell_\infty$ $\epsilon_0$-net (see Definition G.2), for each $q \notin N$, there exists $y \in N$ such that

$$\|y - q\|_\infty \le \epsilon_0 \tag{6}$$

We know

$$|Z(y) - Z(q)| \le L \cdot \|y - q\|_1$$
$$\le L \cdot d\|y - q\|_\infty$$
$$\le L \cdot d\epsilon_0 \tag{7}$$

where the first step follows from Lemma G.7, the second step follows from $\|x\|_1 \le d\|x\|_\infty$ for $x \in \mathbb{R}^d$, and the last step follows from Eq. (6).

Using the on-net query $y$ to answer the off-net query $q$, for any $q \notin N$, we have

$$|f(y) - Z(q)| \le |f(y) - Z(y)| + |Z(q) - Z(y)|$$

$$\leq |f(y) - Z(y)| + L \cdot d \cdot \epsilon_0$$
$$\leq B + L \cdot d \cdot \epsilon_0 \tag{8}$$

where the first step follows from triangular inequality, the second step follows from Eq. (7), the third step follows from Lemma G.6.

Thus, we complete the proof. $\qquad\square$

Therefore, even adaptive queries can be answered accurately, since any adaptive query can be assumed in $\mathcal{B}$.

## H  SOFTMAX ACTIVATION

In this section, we introduce how we extend previous $\ell_p^p$ distance results to the Softmax activation function, which is the most widely used distance measure in attention mechanism based models.

In Section H.1, we show how to extend to the Softmax distance function in Lemma H.6. In Section H.2, we show how to adjust our algorithms. In Section H.3, we extend our algorithm to be robust to adaptive query. In Section H.4, we give the proof of our main result Theorem 3.1.

### H.1  EXPONENTIAL INNER PRODUCT

In this section, we show how we obtain the Softmax distance using $\ell_2^2$ distance query. First, we provide some helpful results from Alman & Song (2023).

**Definition H.1** (Definition 3.1 in Alman & Song (2023)). *Let $r \geq 1$ denote a positive integer. Let $\epsilon \in (0, 0.1)$ denote an accuracy parameter. Given a matrix $A \in \mathbb{R}_{\geq 0}^{n \times n}$, we say $\widetilde{A} \in \mathbb{R}_{\geq 0}^{n \times n}$ is an $(\epsilon, r)$-approximation of $A$ if*

- *$\widetilde{A} = U_1 \cdot U_2^\top$ for some matrices $U_1, U_2 \in \mathbb{R}^{n \times r}$ (i.e., $\widetilde{A}$ has rank at most $r$), and*

- *$|\widetilde{A}_{i,j} - A_{i,j}| \leq \epsilon \cdot A_{i,j}$ for all $(i, j) \in [n]^2$.*

**Lemma H.2** (Lemma 3.4 in Alman & Song (2023)). *Suppose $Q, K \in \mathbb{R}^{n \times d}$, with $\|Q\|_\infty \leq R$, and $\|K\|_\infty \leq R$. Let $A := \exp(QK^\top/d) \in \mathbb{R}^{n \times n}$. For accuracy parameter $\epsilon \in (0, 0.1)$, there is a positive integer $s$ bounded above by*

$$s = O\Big( \max\Big\{ \frac{\log(1/\epsilon)}{\log(\log(1/\epsilon)/R)}, R^2 \Big\} \Big), \tag{9}$$

*and a positive integer $r$ bounded above by*

$$r \leq \binom{2s + 2d}{2s} \tag{10}$$

*such that: There is a matrix $\widetilde{A} \in \mathbb{R}^{n \times n}$ that is an $(\epsilon, r)$-approximation (Definition H.1) of $A \in \mathbb{R}^{n \times n}$. Furthermore, the matrices $U_1$ and $U_2$ defining $\widetilde{A}$ can be computed in $O(n \cdot r)$ time.*

Here we consider the vector version of Lemma H.2.

**Definition H.3.** *We define $\Gamma_{R,s} := \max_{j \in [s]} \frac{R^j}{\sqrt{j!}}$.*

Then, we have $P(x) : [0, R]^d \to [0, \Gamma_{R,s}]^r$ where $P(\cdot)$ is polynomial kernel function defined in Alman & Song (2023).

**Remark H.4.** *We use $\Gamma_{R,s}$ to denote the value range of our polynomial kernel methods function, i.e., $P(x) : [0, R]^d \to [0, \Gamma_{R,s}]^r$. The factorial term in $\Gamma_{R,s}$ comes from Taylor approximation coefficients. We take the maximum overall $s$ order approximation terms to get the upper bound of our value range.*

We use the polynomial approximation method, which has been applied to accelerate Transformer model extensively Alman & Song (2023; 2024a;b); Liang et al. (2024e;b).

**Lemma H.5** (Polynomial approximation). *For any accuracy parameter $\epsilon_s \in (0, 0.1)$, let $R \geq 1$, and let $P(x) : [0, R]^d \to [0, \Gamma_{R,s}]^r$ be the $s$-th order polynomial kernel function defined in Alman & Song (2023) where $r \leq \binom{2s+2d}{2s}$ and $s = O(\max\{\frac{\log(1/\epsilon_s)}{\log(\log(1/\epsilon_s)/R)}, R^2\})$. Then, for any $x, y \in [0, R]^d$, we have*

$$|P(x)^\top P(y) - \exp(x^\top y/d)| \leq \epsilon_s \cdot \min\{\exp(x^\top y/d), P(x)^\top P(y)\}$$

*Furthermore, the vectors $P(x)$ and $P(y)$ can be computed in $O(r)$ time.*

*Proof.* Let $n = 1$. The proof follows from directly applying Lemma H.2. $\square$

Using the results from Alman & Song (2023) above, we can extend our results to Softmax activation.

**Lemma H.6** (Weighted Softmax approximation ). *Let accuracy parameter be $\epsilon_s \in (0, 0.1)$. Let $R \geq 1$. Let $r \leq \binom{2s+2d}{2s}$ and $s = O(\max\{\frac{\log(1/\epsilon_s)}{\log(\log(1/\epsilon_s)/R)}, R^2\})$. Let $P(x) : [0, R]^d \to [0, \Gamma_{R,s}]^r$ be the $s$-th order polynomial kernel function defined in Lemma H.5. Then we can approximate exponential inner product using polynomial kernel function:*

$$| -\frac{1}{2} \sum_{j \in [r]} \sum_{i \in [n]} w_i |P(x_i)_j - P(y)_j|^2 + \frac{1}{2} \sum_{i \in [n]} w_i(\|P(x_i)\|_2^2 + \|P(y)\|_2^2) - w^\top \exp(Xy/d)|$$
$$= O(|w^\top \exp(Xy/d) \cdot \epsilon_s|)$$

*Moreover, the vectors $P(\cdot)$ can be computed in $O(r)$ time.*

*Proof.* From Lemma H.5, we can use a polynomial kernel to approximate the Softmax function:

$$|\sum_{i \in [n]} w_i P(x_i)^\top P(y) - w^\top \exp(Xy/d)| = O(|w^\top \exp(Xy/d) \cdot \epsilon_s|).$$

The proof of approximation error and time complexity of constructing $P(\cdot)$ follows from Lemma H.5.

Then, we can show

$$2 \sum_{i \in [n]} w_i P(x_i)^\top P(y) = -\sum_{i \in [n]} w_i \|P(x_i) - P(y)\|_2^2 + \sum_{i \in [n]} w_i(\|P(x_i)\|_2^2 + \|P(y)\|_2^2)$$
$$= -\sum_{j \in [r]} \sum_{i \in [n]} w_i |P(x_i)_j - P(y)_j|^2 + \sum_{i \in [n]} w_i(\|P(x_i)\|_2^2 + \|P(y)\|_2^2)$$

where the first step follows from $\|x - y\|_2^2 = \|x\|_2^2 + \|y\|_2^2 - 2\langle x, y \rangle$, and the second step follows $\|x\|_2^2 = \sum_{j=1}^d |x_j|^2$ for $x \in \mathbb{R}^d$. $\square$

## H.2 Algorithm Modifications

Based on Lemma H.6, we can now extend our DP algorithms to handle Softmax activation. First, we need to construct $P(y)$ and $P(x_i)$ for $i \in [n]$, each costing $O(r)$ time. ~~Then, for the second term in Lemma H.6, i.e. $\frac{1}{2} \sum_{i \in [n]} w_i(\|P(x_i)\|_2^2 + \|P(y)\|_2^2)$, we don't need to add DP noises in it; instead, we calculate this term exactly, preprocess it, and store the results in the algorithm.~~ For the first term, $-\frac{1}{2} \sum_{j \in [r]} \sum_{i \in [n]} w_i |P(x_i)_j - P(y)_j|^2$, we can adjust our high dimensional DP distance query algorithm to solve it. For the second term in Lemma H.6, i.e., $\frac{1}{2} \sum_{i \in [n]} w_i(\|P(x_i)\|_2^2 + \|P(y)\|_2^2)$, it can be expressed as $\frac{1}{2} \sum_{j \in [r]} \sum_{i \in [n]} w_i |P(x_i)_j - 0|^2$ and $\frac{1}{2} \sum_{i \in [n]} w_i(\sum_{j \in [r]} P(y)_j^2)$. The former can be computed using query 0, while the latter can be solved using the precomputed value $\sum_{i \in [n]} w_i$, which can be obtained from the data $\mathbf{1}_n$ and query 0. Thus, we only need to consider the case $p = 2$ in weighted $\ell_p^p$ distance algorithms.

Now we can give our result that can answer Softmax query.

**Theorem H.7** (Softmax query, formal version of Theorem 4.2)**.** *Let $R \geq 1$. Let $r \leq \binom{2s+2d}{2s}$ and $s = O(\max\{\frac{\log(1/\epsilon_s)}{\log(\log(1/\epsilon_s)/R)}, R^2\})$. Let $\Gamma_{R,s}$ be defined in Definition H.3. Let accuracy parameter be $\epsilon_s \in (0, 0.1)$. There is a data structure* DPTREESOFTMAX *(Algorithm 3) that uses $O(nr)$ spaces to solve Softmax query problem for dataset $X \subset [0, R]^d$ and support the following operations:*

- INIT($X \subset [0, R]^d, n \in \mathbb{N}_+, w \in [-R_w, R_w]^n, \epsilon \in (0, 1), \delta \in (0, 1), \delta' \in (0, 1), c \in (0, 0.1), \epsilon_s \in (0, 0.1)$). *(Algorithm 3) It takes $O(nr)$ time to initialize the data structure.*

- DISTANCEQUERY($y \in [0, R]^d$). *(Algorithm 3) It takes $O(r \log n)$ time to output a number $z$ such that*

  - *the process of output $z$ satisfies $(\epsilon, \delta + \delta')$-DP private, which computes $w^\top \exp(Xy/d)$,*
  - $|z - w^\top \exp(Xy/d)| \leq |\epsilon_s \cdot w^\top \exp(Xy/d)| + O(\epsilon^{-1}\Gamma_{R,s}^2 R_w r \sqrt{\log(1/\delta')} \cdot \log^{3/2} n)$,
  - *it holds with probability* 0.99.

*Proof.* Let $P_{wx} := \sum_{i \in [n]} w_i \|P(x_i)\|_2^2$ and $s_w := \sum_{i \in [n]} w_i$. Observe that $P_{wx} = \sum_{i \in [n]} w_i \|P(x_i) - 0\|_2^2$, meaning we can calculating $P_{wx}$ using query 0. Similarly, $s_w = \sum_{i \in [n]} w_i \|\mathbf{1}_n - 0\|_2^2$, meaning we can calculating $s_w$ using data $\mathbf{1}_n$ and query 0. Thus, we compute $P_{wx}, s_w$ in Line 19 and 22 in Algorithm 3 in this way.

From the privacy proof of Lemma F.1 and the way we choose privacy parameters, similarly we get the output process of calculating $P_{wx}$ and Value is $(\epsilon/3, \delta/3 + \delta'/2)$-DP. Also, the output process of calculating $s_w$ is $(\epsilon/3, \delta/3)$-DP. Then, by Fact B.8, overall process is $(\epsilon, \delta + \delta')$-DP in Line 31 of Algorithm 3.

We then show the time complexity. From Lemma H.6, we know that constructing $P(\cdot)$ requires $O(r)$ time. In the first for loop of INIT, the dominating time consumption is $O(nr)$. The second for loop also has a time complexity of $O(nr)$. Therefore, the total time complexity for INIT is $O(nr)$. In the DISTANCEQUERY function, constructing $P(y)$ takes $O(r)$ time. Within the for loop, it requires $O(r \log n)$. Thus, the total time complexity for DISTANCEQUERY is $O(r \log n)$.

The space complexity is $O(nr)$, since storing the $n \times r$ matrix $P$ is the dominating factor.

The proof of the error follows from the triangle inequality by combining the errors in Lemma H.6 and Theorem F.3. Here, we omit the constant factors of 2 and 3 used for the privacy guarantee in Algorithm 3, incorporating it into the big-$O$ notation for the error analysis. To be more specific, in Line 31 of Algorithm 3, we have 3 terms to bound the error, namely $P_{wx}, s_w \|P(y)\|_2^2$ and Value. From Lemma H.6, the first source of error comes from the approximation error introduced by polynomial kernel method, i.e.,

$$|w^\top \exp(Xy/d) - \frac{1}{2}(\underbrace{\sum_{i \in [n]} w_i \|P(x_i)\|_2^2}_{P_{wx}} + \underbrace{\sum_{i \in [n]} w_i \|P(y)\|_2^2}_{s_w} - \underbrace{\sum_{i \in [n]} w_i \|P(x_i) - P(y)\|_2^2}_{\text{Value}})|$$

$$= O(|\epsilon_s \cdot w^\top \exp(Xy/d)|).$$

Then, the second source of error comes from the DP noises in Theorem F.3, where we use Algorithm 4 to compute the three terms.

The two terms $P_{wx}$ and Value have additive error $O(\epsilon^{-1}\Gamma_{R,s}^2 R_w r \sqrt{\log(1/\delta')} \cdot \log^{3/2} n)$ (Theorem F.3) due to to the way we choose the DP parameters, the application of advanced composition (Theorem B.10), and the transformation of the value range from $[0, R]$ to $[0, \Gamma_{R,s}]$ by the polynomial kernel. See more details in the proof of Lemma F.2.

As for the term $s_w \|P(y)\|_2^2$, the addtive error of $s_w$ is $O(\epsilon^{-1} R_w \log^{3/2} n)$. But since $\|P(y)\|_2^2 \leq r\Gamma_{R,s}^2$, we have the addtive error is $O(\epsilon^{-1}\Gamma_{R,s}^2 R_w r \log^{3/2} n)$ which is smaller than other two terms. We ignore the constant 3 introduced by summing three terms by triangle inequality of absolute function, i.e., $|-t_1 + t_2 + t_3| \leq |t_1| + |t_2| + |t_3|$.

Finally, summing the two sources of error by triangle inequality, we finish the proof. $\square$

## H.3 ADAPTIVE SOFTMAX

In this section, we show how to make Algorithm 3 robust to adaptive query. We follow the same idea from Section G. We notice that, in the Softmax activation, we have query function $Z(y) := w^\top \exp(Xy/d)$ different from the $\ell_1$ distance in Section G. Therefore, we need to re-calculate Lipschitz constant first.

**Lemma H.8** (Lipschitz of weighted Softmax). *If the following conditions hold:*

- *Let data set $X \in [0, R]^{n \times d}$, weights $w \in [-R_w, R_w]^n$, query $y \in [0, R]^d$.*

- *Let $Z(y) := w^\top \exp(Xy/d)$.*

- *Let $L = nd^{-1/2}RR_w \exp(R^2)$.*

*Then, we have $Z(y)$ is L-Lipschitz (note that we have $\ell_1$ Lipschitz here).*

*Proof.* We can show

$$
\begin{aligned}
|Z(y) - Z(\widetilde{y})| &= |\sum_{i \in [n]} w_i \exp(x_i^\top y/d) - \sum_{i \in [n]} w_i \exp(x_i^\top \widetilde{y}/d)| \\
&\leq \sum_{i \in [n]} |w_i| \cdot |\exp(x_i^\top y/d) - \exp(x_i^\top \widetilde{y}/d)| \\
&\leq \sum_{i \in [n]} |w_i| \exp(R^2) |x_i^\top y/d - x_i^\top \widetilde{y}/d| \\
&\leq \sum_{i \in [n]} |w_i| \exp(R^2) \|x_i\|_2 \cdot \|y - \widetilde{y}\|_2/d \\
&\leq nR_w \exp(R^2)\sqrt{d}R \cdot \|y - \widetilde{y}\|_2/d \\
&\leq nd^{-1/2}RR_w \exp(R^2)\|y - \widetilde{y}\|_1
\end{aligned}
$$

where the first step follows from definition of $Z(y), Z(\widetilde{y})$, the second step follows from triangular inequality, the third step follows from Fact B.4, the fourth step follows from Cauchy–Schwarz inequality $|u^\top v| \leq \|u\|_2 \cdot \|v\|_2$ for $u, v \in \mathbb{R}^d$, the fifth step follows from $w_i \in [-R_w, R_w]$ and $x_i \in [0, R]^d$, and the last step follows from $\|u\|_2 \leq \|u\|_1$ for $u \in \mathbb{R}^d$. □

Then we can show how to extend our algorithm to be robust to adaptive query.

**Lemma H.9** (Adaptive Softmax ). *If the following conditions hold:*

- *Let $N$ be the $\ell_\infty$ $\epsilon_0$-net of $\mathcal{B}$, and $|N|$ be the size of net $N$.*

- *Let data set $X \in [0, R]^{n \times d}$, weights $w \in [-R_w, R_w]^n$, query $y \in [0, R]^d$.*

- *Let the failure probability $p_f \in (0, 0.01)$.*

- *We create $l = O(\log((R/\epsilon_0)^r/p_f))$ independent copies of data structure $\{\text{DPTREESOFTMAX}_j\}_{j=1}^l$ (Algorithm 3) and take the median of the outputs with each data structure instantiated with $(\epsilon/l, (\delta + \delta')/l)$-DP.*

- *Let $f(y) := \text{Median}(\{\text{DPTREESOFTMAX}_j.\text{DISTANCEQUERY}(y)\}_{j=1}^l)$.*

- *Let $Z(y) := w^\top \exp(Xy/d)$, where $Z(y)$ is L-Lipschitz with $L = nd^{-1/2}RR_w \exp(R^2)$.*

- *Let $B = O(\epsilon^{-1}l\Gamma_{R,s}^2 R_w r \sqrt{\log(l/\delta')} \cdot \log^{3/2} n)$.*

*Then with probability $1 - p_f$, for all query points $q \in \mathcal{B}$, there exists a point $y \in N$ which is the closest to $q$, we can have the process of outputting the median of $l$ responses is $(\epsilon, \delta + \delta')$-DP and the error satisfies*

$$
|f(y) - Z(q)| \leq |\epsilon_s Z(q)| + B + O(n\sqrt{d}RR_w \exp(R^2)\epsilon_0).
$$

*Proof.* The proof follows from the same idea as the proof of Lemma G.8, except that we use Theorem H.7 and the Lipschitz in Lemma H.8. □

---

**Algorithm 7** Adaptive query data structure

1: **datastructure** DPTREESOFTMAXADAPTIVE             ▷ Theorem 4.4
2: **members**
3:      $\mathcal{D}_1, \ldots, \mathcal{D}_{O(r \log(dR/(\epsilon_s p_f)))}$ : DPTREESOFTMAX       ▷ Algorithm 3
4: **end members**
5: **procedure** INIT($X \subset [0, R]^d, n \in \mathbb{N}_+, w \in [-R_w, R_w]^n, \epsilon \in (0, 1), \delta \in (0, 1), \delta' \in (0, 1), c \in (0, 0.1)), \epsilon_s \in (0, 0.1), p_f \in (0, 0.01))$
6:      $l \leftarrow O(r \log(dR/(\epsilon_s p_f)))$
7:      **for** $i = 1 \rightarrow l$ **do**
8:          $\mathcal{D}_i.\text{INIT}(X, n, w, \epsilon/l, \delta/l, \delta'/l, c, \epsilon_s)$
9:      **end for**
10: **end procedure**
11: **procedure** DISTANCEQUERY($y \in [0, R]^d$)
12:      $l \leftarrow O(r \log(dR/(\epsilon_s p_f)))$
13:      $r \leftarrow 0^l$
14:      **for** $i = 1 \rightarrow l$ **do**
15:          $r_i \leftarrow \mathcal{D}_i.\text{DISTANCEQUERY}(y)$
16:      **end for**
17:      **return** Median of $r$
18: **end procedure**
19: **end datastructure**

---

**Theorem H.10** (Adaptive query Softmax data structure, formal version of Theorem 4.4)**.** *Let $R \geq 1$. Let $r \leq \binom{2s+2d}{2s}$ and $s = O(\max\{\frac{\log(1/\epsilon_s)}{\log(\log(1/\epsilon_s)/R)}, R^2\})$. Let $\Gamma_{R,s}$ be defined in Definition H.3. Let accuracy parameter be $\epsilon_s \in (0, 0.1)$. Let $X \in [0, R]^{n \times d}$ be the dataset, $w \in [-R_w, R_w]^n$ be weights, $y \in [0, R]^d$ be the query, and $p_f$ be the failure probability parameter. Let $l = O(r \log(dR/(\epsilon_s p_f)))$. There is a data structure DPTREESOFTMAXADAPTIVE (Algorithm 7) that uses $O(lnr)$ spaces to solve weighted Softmax query problem for dataset $X \subset [0, R]^d$ and support the following operations:*

- INIT($X \subset [0, R]^d, n \in \mathbb{N}_+, w \in [-R_w, R_w]^n, \epsilon \in (0, 1), \delta \in (0, 1), \delta' \in (0, 1), c \in (0, 0.1), \epsilon_s \in (0, 0.1), p_f \in (0, 0.01)$). *(Algorithm 7) It takes $O(lnr)$ time to initialize the data structure.*

- DISTANCEQUERY($y \in [0, R]^d$). *(Algorithm 7) It takes $O(lr \log n)$ time to output a number $z$ such that*

   - *the process of output $z$ satisfies $(\epsilon, \delta + \delta')$-DP private, which computes $w^\top \exp(Xy/d)$,*
   - $|z - w^\top \exp(Xy/d)| \leq |\epsilon_s \cdot w^\top \exp(Xy/d)| + O(\epsilon^{-1} l \Gamma_{R,s}^2 R_w r \sqrt{\log(l/\delta')} \cdot \log^{3/2} n)$,
   - *it holds with probability $1 - p_f$ (where $p_f$ is used in $l$),*
   - *it is robust to adaptive query.*

*Proof.* We only need to show how to pick $\epsilon_0$ in the parameter $l$, because everything else is the same as Lemma H.9. We know the additive error introduced by adaptive query is $E_a := O(n\sqrt{d}RR_w \exp(R^2)\epsilon_0)$ and the relative error introduced by polynomial kernel approximation is $E_p := w^\top \exp(Xy/d) \cdot \epsilon_s$. It can be shown that:

$$E_p := w^\top \exp(Xy/d) \cdot \epsilon_s$$
$$\leq \epsilon_s \|w\|_2 \cdot \|\exp(Xy/d)\|_2$$
$$= O(nR_w \epsilon_s \exp(R^2))$$

where the first step follows from definition of $E_p$, the second step follows from Cauchy–Schwarz inequality, and the last step follows from $w \in [-R_w, R_w]^n$, $X \in [0, R]^{n \times d}$, and $y \in [0, R]^d$.

Picking $\epsilon_0 = \Theta(\frac{\epsilon_s}{\sqrt{d}R})$, we can hide the error of adaptive query $E_a$ in $E_p$. Thus, we have

$$l = O(\log((R/\epsilon_0)^r/p_f))$$
$$= O(\log((\sqrt{d}R^2/\epsilon_s)^r/p_f))$$
$$= O(r\log(dR/(\epsilon_s p_f)))$$

where the first step comes from the definition of $l$, the second step comes from picking $\epsilon_0 = \Theta(\frac{\epsilon_s}{\sqrt{d}R})$, and the last step follows from $\log(a^d/b) = O(d\log(a/b))$ for any $a > 1, 0 < b < 1, d > 1$. $\qquad\square$

### H.4 PROOF OF MAIN RESULT

In this section, we give the proof of our main result of Theorem 3.1.

**Theorem H.11** (Softmax cross-attention, formal version of Theorem 3.1). *Let $Q, K, V, \text{Attn}$ be defined in Definition 1.1. Assume the input context length $n$ is large enough. Let $p_f$ be the probability of failure parameter. Let $r, s, \epsilon_s$ be parameters of polynomial kernel methods (Lemma H.6). Let $\Gamma_{R,s} := \max_{j\in[s]}\frac{R^j}{\sqrt{j!}}$ (Definition H.3). Let $l = O(r\log(dR/(\epsilon_s p_f)))$. There is a data structure* DPTREECROSSATTENTION *(Algorithm 1) that uses $O(lnrd)$ spaces to ensure cross-attention DP and supports the following operations:*

- INIT$(K, V, \epsilon \in (0,1), \delta \in (0,1), \delta' \in (0,1), c \in (0,0.1), \epsilon_s \in (0,0.1), p_f \in (0,0.01))$ *(Algorithm 1). It takes $O(lnrd)$ time to initialize.*

- *At query time, for user input $Q$, we process one token at a time by passing the $i$-th row of $Q$, denoted $Q_i \in [0,R]^d$, to QUERY$(Q_i)$ (Algorithm 1) for each $i \in [m]$. It takes $O(ldr\log n)$ time to output an entry $z$ in $\text{Attn}(Q, K, V)$ such that*

  - *the process of output $z$ satisfies $(\epsilon, \delta + \delta')$-DP,*
  - *the process of output $z$ has relative error $2\epsilon_s/(1 - \epsilon_s)$,*
  - *the process of output $z$ has additive error $O((1-\epsilon_s)^{-1}n^{-1}\epsilon^{-1}l\Gamma_{R,s}^2 R_w r\sqrt{\log(l/\delta')} \cdot \log^{3/2} n)$,*
  - *it holds with probability $1 - p_f$ (where $p_f$ is used in $l$),*
  - *it is robust to adaptive query.*

*Proof.* We first prove the privacy and then prove error for each coordinate of the output $O$ of Algorithm 1.

**Proof of Privacy:**

From Theorem H.10, $\mathcal{D}_k$.DISTANCEQUERY for $k \in \{0, 1, \ldots, d\}$ in Algorithm 1 answer $(\epsilon/2, \delta/2 + \delta'/2)$-DP queries that are robust to adaptive queries. By Fact B.8, the procedure for calculating each coordinate of vector $O$ is $(\epsilon, \delta + \delta')$-DP in Line 15 of Algorithm 1.

**Proof of Error:**

We prove the error bound of the cross-attention module. We omit the constant factor of 2 used for the privacy guarantee in Algorithm 1, incorporating it into the big-$O$ notation for the error analysis. Let $AV$ be the true value and $\widetilde{AV}$ be the noisy value. Let $D$ be the true value and $\widetilde{D}$ be the noisy value. First, we use triangular inequality to decompose the error:

$$|(D^{-1}AV)_{i,k} - (\widetilde{D}^{-1}\widetilde{AV})_{i,k}|$$
$$\leq |(D^{-1}AV)_{i,k} - (D^{-1}\widetilde{AV})_{i,k}| + |(D^{-1}\widetilde{AV})_{i,k} - (\widetilde{D}^{-1}\widetilde{AV})_{i,k}| \qquad (11)$$

We now prove for each term.

**Part 1: Error bound for $AV$**

From Section 3, we know that we can ensure matrix $AV$ in cross-attention computation satisfies DP. Next, from Theorem 4.4, for $i \in [m], j \in [n], k \in [d]$, we have $(AV)_{i,k}$ is $(\epsilon, \delta + \delta')$-DP and also robust to adaptive query.

Let $\zeta := \epsilon^{-1} l \Gamma_{R,s}^2 R_w r \sqrt{\log(l/\delta')} \cdot \log^{3/2} n$ denote the additive error. Then, from Theorem H.10, we have

$$|(AV)_{i,k} - \widetilde{(AV)}_{i,k}| \le |\epsilon_s \cdot (AV)_{i,k}| + O(\zeta) \tag{12}$$

For $D_{i,i}$, we can show

$$D_{i,i} = (A \cdot \mathbf{1}_n)_i = \sum_{j=1}^{n} \exp(\langle Q_i, K_j \rangle / d) \ge n \tag{13}$$

because $\langle Q_i, K_j \rangle \ge 0$ for bounded $Q, K$.

Finally, we can show the error of first term in Eq. (11) is bounded by

$$|(D^{-1} AV)_{i,k} - (D^{-1} \widetilde{AV})_{i,k}| = |D_{i,i}^{-1} ((AV)_{i,k} - \widetilde{(AV)}_{i,k})|$$
$$= |D_{i,i}^{-1}| \cdot |((AV)_{i,k} - \widetilde{(AV)}_{i,k})|$$
$$\le |\epsilon_s \cdot D_{i,i}^{-1} (AV)_{i,k}| + O(n^{-1}\zeta)$$

where the first step follows from definition, the second step follows from simple algebra, and the last step follows from Eq. (12) and (13).

**Part 2: Error bound for $D$**

We initialize one DPTREESOFTMAXADAPTIVE $\mathcal{D}_0$ with INIT$(K, n, \mathbf{1}_n, \epsilon, \delta, \delta', c, \epsilon_s, p_f)$ in Algorithm 1 to compute $D$. Notice that we input $\mathbf{1}_n$ as the third argument.

Recall that

$$D_{i,i} = \sum_{i=1}^{n} \exp(\langle Q_i, K_j \rangle / d)).$$

This can be viewed as the weighted Softmax problem but with weight $\mathbf{1}_n$. To be more clear, let us recall that $R_w$ is the upper bound of the entries in $V$, and define $R_w'$ as the upper bound of the entries in $\mathbf{1}_n$. Observe that we can reuse previous results in Theorem H.10 with adjustment only on the value of $R_w'$ (which is 1) in $\mathcal{D}_0$.

We wish to bound

$$|D_{i,i}^{-1} - \widetilde{D}_{i,i}^{-1}| = \frac{|D_{i,i} - \widetilde{D}_{i,i}|}{D_{i,i} \cdot \widetilde{D}_{i,i}}.$$

For the term $|D_{i,i} - \widetilde{D}_{i,i}|$, similar to Eq. (12), from Theorem H.10, we have

$$|D_{i,i} - \widetilde{D}_{i,i}| \le |\epsilon_s \cdot D_{i,i}| + O(\zeta), \tag{14}$$

where we assume $R_w \ge 1 = R_w'$ and loose the $R_w'$ in additive error parameter in $\mathcal{D}_0$ from 1 to $R_w$.

Now we need the lower bound of $\widetilde{D}_{i,i}$. From Eq. (14), we have

$$\widetilde{D}_{i,i} \ge D_{i,i} - (|\epsilon_s \cdot D_{i,i}| + O(\zeta)) \ge |(1 - \epsilon_s) \cdot D_{i,i}| - O(\zeta).$$

Then, we have

$$|D_{i,i}^{-1} - \widetilde{D}_{i,i}^{-1}| = D_{i,i}^{-1} \frac{|D_{i,i} - \widetilde{D}_{i,i}|}{\widetilde{D}_{i,i}} \le D_{i,i}^{-1} \frac{|\epsilon_s \cdot D_{i,i}| + O(\zeta)}{|(1 - \epsilon_s) \cdot D_{i,i}| - O(\zeta)}$$

We assume $n$ is large enough and thus ignore other small factors. Observe that $O(\zeta) = O(\log^{3/2} n)$, and $D_{i,i} \ge n = O(n)$ from **Part 1**. Thus, $O(\zeta)$ is a small order term compared to $D_{i,i}$. As a consequence, we get

$$|D_{i,i}^{-1} - \widetilde{D}_{i,i}^{-1}| \le D_{i,i}^{-1} \frac{|\epsilon_s \cdot D_{i,i}|}{|(1 - \epsilon_s) \cdot D_{i,i}|} = D_{i,i}^{-1} \frac{\epsilon_s}{(1 - \epsilon_s)}, \tag{15}$$

since $\epsilon_s \in (0, 0.1)$.

From Eq. (12), we have

$$|\widetilde{(AV)}_{i,k}| \leq (1 + \epsilon_s) \cdot |(AV)_{i,k}| + O(\zeta)$$

We consider the second term in Eq.(11). Then,

$$|(D^{-1}\widetilde{AV})_{i,k} - (\widetilde{D}^{-1}\widetilde{AV})_{i,k}|$$

$$= |D_{i,i}^{-1} - \widetilde{D}_{i,i}^{-1}| \cdot |\widetilde{(AV)}_{i,k}|$$

$$\leq D_{i,i}^{-1} \frac{\epsilon_s}{(1 - \epsilon_s)}((1 + \epsilon_s) \cdot |(AV)_{i,k}| + O(\zeta))$$

$$= \epsilon_s \frac{(1 + \epsilon_s)}{(1 - \epsilon_s)} \cdot D_{i,i}^{-1}|(AV)_{i,k}| + O(\frac{\epsilon_s}{(1 - \epsilon_s)}D_{i,i}^{-1}\zeta)$$

$$\leq \epsilon_s \frac{(1 + \epsilon_s)}{(1 - \epsilon_s)} \cdot |D_{i,i}^{-1}(AV)_{i,k}| + O(\frac{\epsilon_s}{(1 - \epsilon_s)}n^{-1}\zeta)$$

where the first step follows from simple algebra, the second step follows from the previous derived upper bounds, the third step follows from simple algebra, and the last step follows from Eq.(13).

**Part 3: Final error bound**

Combining results from **Part 1 and 2**, the final error bound is

$$|(D^{-1}AV)_{i,k} - (\widetilde{D}^{-1}\widetilde{AV})_{i,k}|$$

$$\leq |(D^{-1}AV)_{i,k} - (D^{-1}\widetilde{AV})_{i,k}| + |(D^{-1}\widetilde{AV})_{i,k} - (\widetilde{D}^{-1}\widetilde{AV})_{i,k}|$$

$$= \epsilon_s \cdot |D_{i,i}^{-1}(AV)_{i,k}| + O(n^{-1}\zeta) + \epsilon_s \frac{(1 + \epsilon_s)}{(1 - \epsilon_s)} \cdot |D_{i,i}^{-1}(AV)_{i,k}| + O(\frac{\epsilon_s}{(1 - \epsilon_s)}n^{-1}\zeta)$$

$$= \frac{2\epsilon_s}{(1 - \epsilon_s)} \cdot |(D^{-1}AV)_{i,k}| + O((1 - \epsilon_s)^{-1}n^{-1}\zeta)$$

Therefore, we prove the error bound.

$\square$

