# OpenReview forum: "Differential Privacy of Cross-Attention with Provable Guarantee"
_ICLR.cc/2025/Conference — Submitted to ICLR 2025_

### Official Review · Reviewer_MwpB · 2024-10-28

**Soundness:** 3
**Presentation:** 3
**Contribution:** 2
**Rating:** 6
**Confidence:** 3

**Summary:**

This paper studies the problem of privately computing cross-attention. The authors propose a data structure that can answer cross-attention queries within certain relative and absolute errors while satisfying $(\varepsilon,\delta)$-DP. The main idea is to approximate the exponential inner product using a polynomial kernel function, which converts the original problem into one of answering distance queries.

**Strengths:**

The paper is the first to propose a DP algorithm for cross-attention computation. The algorithm is both time- and space-efficient for initialization and can quickly answer adaptive user queries. The idea of approximating the exponential inner product by a polynomial kernel function (Lemma I.5) and the distance query algorithms (Algorithms 5 and 6) are interesting.

**Weaknesses:**

1. It is unclear what information is protected in this paper.  It is necessary to clearly state which parts of the input are private ($K$ and $V$?) and which parts of the output are public ($Q$?). Moreover, since each row of $K$ is derived from one word in the input, the proposed algorithm seems to only protect word-level privacy.
2. The paper does not contain any experiments, making it difficult to evaluate whether the obtained errors fall within a reasonable range. Since the problem arises from a real-world application, it is strongly recommended to provide some illustrative experiments, such as standard NLP tasks on real-world datasets.
3. The proposed algorithm has a relative error, which could be unsatisfactory. In Section 5, it is mentioned that the parameter $\alpha$ can be removed. Does this imply that the algorithm can be modified to involve only additive errors, or does the $n^{-1}\varepsilon_{S}$ term still exist?

**Questions:**

1. The definition of differential privacy in the article is unclear. It is defined for datasets represented by vectors in $\mathbb{R}^n$. However, in cross-attention, the key and value matrices have dimensions of $n\times d$. I recommend clarifying the definition so that it fits the cross-attention setting.
2. In line 269, it is stated that "D is only a normalization factor and does not contain sensitive information". However, the normalization factors depend on $A$, which is calculated from $K$ and $V$. Thus, $D$ contains private information and should be protected. I do not understand why we can directly discard $D$ in this context.
3. In Algorithm 3, the variables $P_{wx}$ and $s_{wx}$ are used without adding any noise. Can you explain why the resulting algorithm still preserves differential privacy?
4. Has the problem of private distance queries been studied in the literature? If so, it would be helpful to include some related work.
5. The initialization time complexity and query time complexity in the abstract do not include $d$. Is this a typo?

---

> ### Author Response · Authors · 2024-11-16
> **Official Comment by Authors (Part 1)**
>
> We thank the reviewer for their valuable suggestions. We provide our response below and hope it will address your concerns.
>
> ### W1.1: It is unclear what information is protected in this paper. It is necessary to clearly state which parts of the input are private (K and V?) and which parts of the output are public (Q?).
>
> K and V are values stored in the memory and disks, and Q is obtained only when the user types its input. Thus, K and V are private assets protected in the model, and Q is public. We want to emphasize that we ensure the whole cross-attention process is DP ensuring privacy guarantee. We revised our paper to add this discussion in Line 223-226.
>
> ### W1.2: Moreover, since each row of K is derived from one word in the input, the proposed algorithm seems to only protect word-level privacy.
>
> We thank the reviewer’s valuable comments. We agree that our neighbor dataset is defined based on word-level so that we protect word-level privacy. Our methods can still provide some sentence-level privacy as we protect every word, e.g., every token in a sentence may be perturbed. How to build a data structure designed for sentence-level privacy protection is an interesting and challenging topic. One thought is to use the notion of group privacy in [1] which states the neighbor dataset can differ in multiple entries. We are willing to discuss more per the reviewer’s request.
>
> ### W2: The paper does not contain any experiments, making it difficult to evaluate whether the obtained errors fall within a reasonable range.
>
> Thank you for the thoughtful feedback. We acknowledge that empirical validation can offer valuable insights into real-world performance; however, implementing our theoretical framework would require significant engineering effort to address system-level challenges, such as developing distributed algorithms and efficiently implementing polynomial kernel methods on GPUs. These challenges fall beyond the scope of our theoretical contribution.
>
> Our primary contribution lies in providing the first mathematically rigorous privacy guarantees for cross-attention, with provable bounds on error, computational complexity, and memory requirements. We aimed to address the fundamental question: "Is it possible to make cross-attention differentially private with provable guarantees?" Our work answers this affirmatively, offering concrete mathematical bounds. We believe is a significant contribution in itself, which is also supported by Reviewer s8of that “the paper is already in the current state fairly dense.”
>
> ### W3: Does this imply that the algorithm can be modified to involve only additive errors, or does the $n^{-1}\epsilon_s$ term still exist?
>
> After removing $\alpha$, relative error $n^{-1} \epsilon_s$ would still exist since it is introduced by the polynomial kernel method in Section I Lemma I.5. Also if $n$ gets larger, the relative error would be smaller. Besides, relative errors are actually good criteria and many existing works use relative errors, e.g., [3,5,6].
>
> ### Q1: The definition of differential privacy in the article is unclear.
> The $n \times d$ matrix we study can be viewed as a set of $d$ vectors with length $n$. And in our Algorithms, we initialize the $n \times d$ matrix as $d$ times of a dimension $n$ DPTREE data structure, which utilizes $\mathbb{R}^n$ as inputs. Thus, the definition of DP applies to this scenario.
>
> ### Q2: I do not understand why we can directly discard D in this context.
> $D$ depends on $A$, which is calculated using $Q$ and $K$, serving as a normalizing factor that gathers all the information. As discussed in **W1.1**, we consider $K$ and $V$ as private assets. In our data structure, we store both $K$ and the corresponding noises. When computing $AV$, we use the perturbed $K$ to ensure DP guarantees, while for computing $D$, we use the original $K$ without perturbation. By the post-processing property of DP (Fact B.5, Line 1114), the entire cross-attention mechanism remains DP. This is what we mean by $D$ containing no sensitive information. Our current design consequently guarantees an inversely proportional relationship between error and input context length $n$, aligning with the intuition that a larger number of tokens blends the information, enhancing privacy and utility. We have revised the paper to elaborate further on this. We add this discussion in our revised paper Line 229-231.
>
> ### Q3: In Algorithm 3, the variables $P_{wx}$ and $s_{wx}$ are used without adding any noise. Can you explain why the resulting algorithm still preserves differential privacy?
>
> This is because of the post-processing property of DP (Fact B.5, Line 1114), which states in an algorithm that if one step is DP, all the following steps are DP. Since we have the weighted distance term being DP, our whole process is DP.

---

> ### Author Response · Authors · 2024-11-16
> **Official Comment by Authors (Part 2)**
>
> ### Q4: Has the problem of private distance queries been studied in the literature? If so, it would be helpful to include some related work.
>
> Private distance queries have been studied in the literature [2,3,4]. We have updated them in Section 4.2 in the revised version Line 349-354.
>
> ### Q5: The initialization time complexity and query time complexity in the abstract do not include $d$. Is this a typo?
>
> Thank you for your observation. This is not a typo. We use the polynomial kernel method, which maps a $d$-dimension vector to $r$-dimension, where $r \ge d$ by the polynomial method. Thus, our initialization and query time depends on $d$, which is contained in $l=O(r\log(dR/(\epsilon_s p_f)))$ Line 249, and we use $\widetilde{O}$ to hide the log dependence. We have revised the paper to contain a formal definition of $\widetilde{O}$ in Line 154-157.
>
> [1] Dwork, C., & Roth, A. (2014). The algorithmic foundations of differential privacy. Foundations and Trends in Theoretical Computer Science.
>
> [2] Huang, Z., & Roth, A. (2014). Exploiting metric structure for efficient private query release. In Symposium on Discrete Algorithms (SODA’14).
>
> [3] Backurs, A., Lin, Z., Mahabadi, S., Silwal, S., & Tarnawski, J. (2024). Efficiently Computing Similarities to Private Datasets. ICLR’24.
>
> [4] Liu, E., Hu, J. Y. C., Reneau, A., Song, Z., & Liu, H. (2024). Differentially private kernel density estimation. arXiv preprint arXiv:2409.01688.
>
> [5] Song, Z., Woodruff, D. P., & Zhong, P. (2019). Relative error tensor low rank approximation.  In Symposium on Discrete Algorithms (SODA’19).
>
> [6] Wang, S., Gittens, A., & Mahoney, M. W. (2019). Scalable kernel k-means clustering with nystrom approximation: Relative-error bounds. Journal of Machine Learning Research (JMLR’19).

---

> > ### Comment · Reviewer_MwpB · 2024-11-22
> >
> > Thank you for your reply. I still have some questions.
> >
> > 1. Definition of Differential Privacy
> >
> > The current definition in the paper states that two datasets are considered neighboring if the $\ell_1$ distance between them is $1$. This definition is commonly applied in contexts where datasets are represented as histograms. However, I believe it is not appropriate for this article, where two matrices $X, X'\in\mathbb{R}^{n\times d}$ are considered neighboring if they differ by a single row -- in this situation, the $\ell_1$ distance can be as large as $2dR$.
> >
> > I suggest the authors define differential privacy for matrices rather than vectors, as the actual input to the entire algorithm is represented by matrices. Also, extending DP from vectors to matrices requires the composition theorem, which incurs a $\sqrt{d}$ factor (e.g., line 11 in Alg 7) and could potentially confuse readers if not defined properly.
> >
> > 2. Use of Post-processing
> >
> > The post-processing property (Fact B.5) requires the post-processing algorithm ($\mathcal{A}_2$) to be independent of the input data.
> >
> > In the computation of $D^{-1}AV$, $\mathcal{A_1}$ generates a private version of $AV$, while $\mathcal{A}_2$ computes $D$ from the unperturbed matrix $K$. This does not meet the above independence condition, as $\mathcal{A}_2$ relies on the sensitive data matrix $K$, thus compromising privacy.
> >
> > In Alg 3, the variable $\text{Value}$ remains private until line 26. However, $P_{wx}$ and $s_{wx}$ are calculated from the sensitive data without adding any noise. Again, the post-processing property is not applicable here since $\mathcal{A}\_{2}$ is the computation of $P_{wx} + s_{wx}\lVert P(y)\rvert_2^2 - \text{Value}$, which does not the independence requirement.
> >
> > 3. Dependence on $d$
> >
> > Theorem 3.1 indicates that the time to initialize a data structure is $O(lnr)$. Since there are $d$ such structures, I am curious why the complexity is stated as $\widetilde{O}(nr^2)$ in the abstract instead of $\widetilde{O}(ndr^2)$.

---

> > > ### Author Response · Authors · 2024-11-24
> > > **Thank you and further reply to new concerns**
> > >
> > > We are glad that our reply has addressed some concerns from the reviewer! We sincerely thank you for your insightful response. We appreciate your time and we would like to answer your valuable questions.
> > >
> > > ### Q1: Definition of Differential Privacy
> > >
> > > Thank you for your valuable suggestion. We revised our paper’s definition from vector to matrix in Line 166-174. Thanks for your meticulous and insightful comments.
> > >
> > >
> > > ### Q2: Use of Post-processing
> > >
> > > Thanks for your comment! We agree with your comments and we have fixed that problem and updated our abstract, main body, and Appendix I by brown text.
> > >
> > > In our second version revision, we added noises using DPTREE data structures to calculate $P_{wx}$, $s_w$, and $D$, and reanalyze the privacy and utility. We refer the reviewer to our second version revision for more details.
> > >
> > > We will clean our writing and try to improve algorithms further.
> > >
> > >
> > > ### Q3: Dependence on $d$
> > >
> > > Thanks for pointing out the question and for your thorough reading! Very good catch. We revised our abstract in Line 17-26 and main body.
> > >
> > > We hope our answer may address your concern and we are willing to discuss more per reviewer requests.

---

> > > > ### Comment · Reviewer_MwpB · 2024-11-25
> > > >
> > > > Thank you for your efforts in revising the proof. After a quick review, I still have a few questions regarding the proof:
> > > >
> > > > 1. I would appreciate more details in the analysis of Theorem I.8, particularly concerning the error bounds. The algorithm estimates three terms: $t_1 = \sum_{i\in [n]}w_i\lVert P(x_i) - P(y)\rVert_2^2$, $t_2=\sum_{i\in[n]} w_i\lVert P(x_i)\rVert_2^2$, and $t_3 = \sum_{i\in[n]}w_i\lVert P(y)\rVert$. Since the result is $ -t_1 + t_2 + t_3 $ and the terms have different signs, I am unclear how the relative error is controlled using the triangle inequality.
> > > >
> > > > 2. In Alg 1, each $\mathcal{D}_k$ is assigned parameters $\varepsilon / 2, \delta / 2, \delta' / 2$. Should these be $\varepsilon / (d +1), \delta / (d + 1), \delta' / (d + 1)$ instead, to ensure the composition theorem applies? If so, the final error bound may need to be adjusted accordingly.
> > > >
> > > > 3. Could you please elaborate on Part 2 of the proof of Theorem I.12? I find it difficult to follow the reasoning for bounding $K_j - \tilde{K}_j$: it is unclear where $\tilde{K}_j$ comes from. The data structure DPTREESOFTMAXADAPTIVE (Alg 8) looks quite complex. It initiates $l$ instances of Alg 3, which further transforms the input using the polynomial kernel function. I am confused about how this relates to bounding the sum of truncated Laplace noises.
> > > >
> > > > 4. Since the proof has been revised, some text needs to be updated accordingly. For instance, the first paragraph of Section I.2 still states that "we don’t need to add DP noises", while the revised algorithm adds noises.
> > > >
> > > > Thank you for addressing these points.

---

> ### Author Response · Authors · 2024-11-26
> **Thank you and answer to follow-up questions**
>
> We are so glad that you appreciate our revision of the proof. We will answer your follow-up questions. We also upload the third version of our revision for your reference.
>
> ### Q1: Error bound of Theorem I.8
> Thanks for your comment. We revised the proof in Line 2401-2422 for more details about how we control the error bounds.
>
> ### Q2: Privacy parameters of Algorithm 1
> Thanks for your valuable suggestion. We agree that the $d+1$ denominator should be applied. We add a Remark 3.2 in Line 269-273.
>
> ### Q3: How to bound $K_j - \widetilde{K}_j$
> Thank you for your peruse and insightful question! After careful checking, we found a better way to prove our utility bound, and we can get tighter and more elegant results than the second version revision, where we can directly get $|D\_{i,i} - \tilde{D}\_{i,i}|$ by our Theorem. We revised our proof to be more clear and improve our error bound. See Line 2546-2671.
>
> ### Q4: Texts need to update
> Thanks for your careful reading. We revised the paper’s text more. See Line 2326-2333.
>
> We are so happy during the discussion with you. Your suggestion helps us a lot. We are so thankful that you asked those valuable questions. We are willing and glad to improve our draft if the reviewer has more suggestions.

---

> > ### Comment · Reviewer_MwpB · 2024-11-26
> >
> > Thank you for your response. The new proof of Theorem I.12 is elegant and provides better error bounds.
> >
> > I would like to clarify my question regarding Theorem I.8.
> >
> > Using the notations I defined earlier, there are three terms to be estimated, which satisfy the equation $0.5(-t_1 + t_2 + t_3) = w^{\top}\exp(Xy/d)\cdot (1 \pm O(\epsilon_S))$. According to the proof, the algorithm computes $\hat{t}_i = t_i\cdot(1 \pm \alpha) \pm \eta$ for each $i$, where $\eta$ represents the additive error.
> >
> > My concern is that combining these three estimated values gives:
> >
> > $(-\hat{t}_1 + \hat{t}_2 +\hat{t}_3) = (-t_1 + t_2 + t_3) \pm 3\eta \pm \alpha(|t_1| + |t_2| + |t_3|)$.
> >
> > This expression does not yield a relative error of $\alpha$. Could you please clarify this point? Thank you!
> >
> > Additionally, upon reviewing the details regarding the relative error, I noticed that in several bounds (e.g., in Lemma F.5, Theorem 4.4, ...), it is expressed as $|\hat{A} - A_{\*}| \le \alpha A_{\*} + \text{additive error}$. This looks a bit strange since $A_{\*}$ is the weighted distance, which may be negative because the weights can be negative. Should this be $|A_{\*}|$ instead?
> >
> > Thank you for your attention to these matters!

---

> ### Author Response · Authors · 2024-11-28
> **Thank you and further reply to new concerns**
>
> We sincerely thank you for your careful check and valuable questions. Based on your comments, we can improve our analysis further.
>
> ### Q1: Relative error
> We thank the reviewer point this out. We agree with the reviewer that the relative error cannot be combined when a negative term exists, we did not notice that in our previous round rebuttal.
>
> In the fourth version, to fix this issue, we improve our DPTree data structure to remove the relative error term related to $\alpha$. In the original version (before the rebuttal discussion), we introduced the idea of removing the $\alpha$ term in Section 5 Discussion, paragraph “How to remove the relative error parameter $\alpha$?” in Line 503-510. Based on these ideas, in Line 298-308 and 1337-1403, we update our DPTree, such that there is only an additive error in Line 1404-1474. Then, there is no relative error combination issue anymore. We refer the reviewer to the fourth version revision for more details.
>
> ### Q2: Absolute value
> We have fixed these typos in Line 1881-1889 and all other places.
>
> We deeply appreciate that the reviewer helped us improve the paper's results and quality!

---

> > ### Comment · Reviewer_MwpB · 2024-12-01
> >
> > Thank you for your clarification. I think there are no major issues in the current version.
> >
> > I have raised my score accordingly.

---

> > > ### Author Response · Authors · 2024-12-01
> > > **Thank you**
> > >
> > > We sincerely thank the reviewers for the many constructive and helpful comments that helped us improve the draft. We thank you for the score raising! We have a brilliant rebuttal discussion experience. We hope AC may nominate reviewer MwpB as a top reviewer. Best, Authors

---

### Official Review · Reviewer_9PH2 · 2024-11-04

**Soundness:** 3
**Presentation:** 2
**Contribution:** 3
**Rating:** 6
**Confidence:** 3

**Summary:**

This paper proposes a differentially private algorithm for implementing Cross Attention. It reformulates Cross Attention as a private summation problem and employs a classic binary tree algorithm to ensure privacy. The paper provides a comprehensive analysis of the proposed algorithm, though it lacks empirical evaluation.

**Strengths:**

1. The novel idea of reformulating Cross Attention as a weighted distance problem allows the use of classic differential privacy algorithms, such as the binary tree mechanism, as building blocks to ensure the privacy of Cross Attention.

2. A comprehensive analysis of the algorithm is provided, covering aspects such as utility, computational complexity, and privacy properties. I believe this work provides a solid foundation for developing differentially private mechanisms for core components in large language models.

3. Figure 1 in the appendix is very helpful for understanding the connection between cross attention and the dp binary tree.

**Weaknesses:**

1. The comparison with DP-SGD is limited. Currently, DP-SGD is widely used in various fine-tuning tasks and can be relatively easily integrated into existing systems. In contrast, the proposed DP Cross-Attention requires complex algorithmic design, making adaptation to existing systems challenging. It requires significant modifications, which reduces its accessibility.

2. For linear queries, the matrix mechanism [1] generally outperforms the binary tree. I would be interested in hearing more discussion on the matrix mechanism. Could you also provide an estimate of the width of the binary tree (i.e., the number of leaves) in your algorithm?

[1] Li, Chao, Gerome Miklau, Michael Hay, Andrew McGregor, and Vibhor Rastogi. "The matrix mechanism: optimizing linear counting queries under differential privacy." The VLDB journal 24 (2015): 757-781.

**Questions:**

In Theorem 1.2, you mentioned that with probability 1 - $p_f$, the mechanism is $(\epsilon, \delta)$-DP. It gives me a feeling that the mechanism is $(\epsilon, \delta + p_f) $- DP. What's the value of $p_f$ in practice? Given a fixed $\epsilon$, the larger is $ \delta + p_f$ , the less private is the mechanism.

---

> ### Author Response · Authors · 2024-11-16
>
> We thank the reviewer for their valuable suggestions. We provide our response below and hope it will address your concerns.
>
> ### W1: The comparison with DP-SGD is limited.
>
> We acknowledge the success of DP-SGD as the standard approach for applying DP to ML. However, it limits the scope of DP to the optimizer. In contrast, we propose a novel perspective on integrating DP directly into the attention mechanism, supported by strong theoretical analysis and guarantees. Given the resource-intensive nature of LGM training, our technique offers a practical alternative for models trained with standard SGD, which lack privacy guarantees. In such cases, DP-SGD would require retraining the models, which is computationally expensive, while our method does not. Exploring the practicality of our method and improving its efficiency will be our focus for future work. We added the DP-SGD discussion in the revised version Line 124-129.
>
> ### W2.1: For linear queries, the matrix mechanism [1] generally outperforms the binary tree. I would be interested in hearing more discussion on the matrix mechanism.
>
> Thanks for your insightful suggestion. For the matrix mechanism [1], a preliminary thought would be: Consider $A = \exp (Q K^\top)$ defined in Line 74. The $Q$ of size $m \times d$ can be viewed as the query matrix with $m$ linear queries in Definition 4 of [1] and $K$ is then the database. Then we could use the results from [1] to design an alternative algorithm to improve the current binary tree data structure DPTREE. We added the corresponding discussion in the revised version Line 484-488.
>
> ### W2.2: Could you also provide an estimate of the width of the binary tree (i.e., the number of leaves) in your algorithm?
>
> Our method requires $n$ number of leave nodes for one DPTREE data structure in Algorithm 2 Line 288.
>
> ### Q1.1: It gives me a feeling that the mechanism is $(\epsilon,\delta+p_f)$-DP.
>
> The mechanism is indeed $(\epsilon,\delta+\delta')$-DP as proved.
> - The $\delta$ is DP failure probability.
> - The $\delta’$ controls a hidden constant term as discussed in Remark 4.3 Line 439.
> - The failure probability $p_f$ is to control the success probability of the adaptive query error control detailed in Section H, which boosts the constant probability to arbitrary high probability.
>
> Though with a similar name, the failure probability parameter $p_f$ is not related to or participated in the DP failure probability $\delta$.
>
> ### Q1.2: What's the value of $p_f$ in practice?
>
> In practice, $p_f$ can be set to 0.001 to ensure a high success rate of 0.999.
>
> [1] Li, C., Miklau, G., Hay, M., McGregor, A., & Rastogi, V. (2015). The matrix mechanism: optimizing linear counting queries under differential privacy. The VLDB journal, 24, 757-781.

---

### Official Review · Reviewer_s8oF · 2024-11-04

**Soundness:** 3
**Presentation:** 2
**Contribution:** 3
**Rating:** 6
**Confidence:** 2

**Summary:**

This paper introduces a modified cross attention algorithm with differential privacy guarantees. The authors modify the cross-attention algorithm by approximating the exponential inner product which allows them to cast the modified algorithm into a weighted distance problem. The paper introduces a several novel tree based data structures to efficiently compute a private the private weighted distance. They produce detailed proofs of privacy guarantees as well as error analysis.

**Strengths:**

- Given that cross-attention modules are fundamental in numerous applications involving sensitive data such as RAG, this work addresses an important gap. The transformation of cross-attention computations into a "weighted distance problem" is a creative solution, making it amenable to DP techniques in a novel way.
- The paper does a good job of grounding its methodology in formal mathematical guarantees. By presenting detailed proofs of differential privacy for the cross-attention mechanism as well as an error and complexity analysis, the authors provide a robust theoretical backing.

**Weaknesses:**

- The appeal of cross attention is its superior empirical performance. The paper adapts the algorithm in several ways without evidence how the algorithm's performance is changed.
- While I understand that the contributions of this paper are of theoretical nature, the motivation of the paper is the popularity of the cross-attention mechanism which is due to its practicality. As such, an empirical evaluation of the utility-privacy trade-off would strengthen the paper especially considering the main audience of this conference. However, the paper is already in the current state fairly dense and I'm not sure whether adding even more content will help the readability in its current format.

**Questions:**

- How do you guarantee that $K_{i,j} \in [0,R]$ and $V_{i,j} \in [-R,R]$? Is this done by clipping? If so, what effect does that have on the performance of the algorithm?
- What is the scope of the adjacency relation? Is it leaving out individual tokens? If so, in practice secrets may reach across many tokens and the DP guarantees would not provide meaningful privacy.

---

> ### Author Response · Authors · 2024-11-16
>
> We thank the reviewer for their valuable suggestions. We provide our response below and hope it will address your concerns.
>
> ### W1 & W2: The appeal of cross attention is its superior empirical performance. The paper adapts the algorithm in several ways without evidence of how the algorithm's performance is changed.
>
> Thank you for the thoughtful feedback. We theoretically show how our method’s error and time and memory cost change. We acknowledge that empirical validation can offer valuable insights into real-world performance; however, implementing our theoretical framework would require significant engineering effort to address system-level challenges, such as developing distributed algorithms and efficiently implementing polynomial kernel methods on GPUs. These challenges fall beyond the scope of our theoretical contribution.
>
> Our primary contribution lies in providing the first mathematically rigorous privacy guarantees for cross-attention, with provable bounds on error, computational complexity, and memory requirements. We aimed to address the fundamental question: "Is it possible to make cross-attention differentially private with provable guarantees?" Our work answers this affirmatively, offering concrete mathematical bounds. We believe this is a significant contribution in itself, which is also supported by your comments that “the paper is already in the current state fairly dense.”
>
> ### Q1: How do you guarantee that $K_{i,j} \in [0,R]$ and $V_{i,j} \in [−R,R]$? Is this done by clipping? If so, what effect does that have on the performance of the algorithm?
>
> Yes, we could use clipping or quantization techniques to ensure the range of $K$ and $V$. Also, we can view $R$ as the maximum absolute value of the $K$ and $V$ entries. We assume in advance that the range is clipped for theoretical proof purposes.
>
> ### Q2: What is the scope of the adjacency relation? Is it leaving out individual tokens? If so, in practice secrets may reach across many tokens and the DP guarantees would not provide meaningful privacy.
>
> We thank the reviewer’s valuable comments. We agree that our neighbor dataset is defined based on word-level so that we protect individual token privacy. Our methods can still provide some sentence-level or many tokens privacy as we protect every word, e.g., every token in a sentence may be perturbed. How to build a data structure designed for sentence-level privacy protection is an interesting and challenging topic. One thought is to use the notion of group privacy in [1], which states that the neighbor dataset can differ in multiple tokens. We are willing to discuss this further per the reviewer’s request.
>
> [1] Dwork, C., & Roth, A. (2014). The algorithmic foundations of differential privacy. Foundations and Trends in Theoretical Computer Science.

---

> > ### Comment · Reviewer_s8oF · 2024-11-26
> >
> > Thank you for the detailed responses.
> >
> > Regarding Q1, I don't quite follow how we could view $R$ as the maximum value of $K$ and $V$. Aren't $K$ and $V$ dataset dependent. The sensitivity is defined quantifying over all pairs of neighbouring datasets so also datasets with very large values $R$. A similar problem exists in DP-SGD where it is solved by clipping the values to predefined fixed range.
> >
> > Regarding Q2, so this would lead to a DP $\varepsilon$ that scales with the sequence length? Using group privacy even moderate sequence lengths of 100 tokens or more significantly reduces the privacy protection.

---

> > > ### Author Response · Authors · 2024-11-28
> > > **Thank you and further reply to new concerns**
> > >
> > > We sincerely thank to your your response and your time. We answer your questions below.
> > >
> > > ### Q1.1:  How we could view $R$ as the maximum value of $K$ and $V$. Aren't $K$ and $V$ dataset dependent. The sensitivity is defined quantifying over all pairs of neighbouring datasets so also datasets with very large values $R$. The sensitivity is defined quantifying over all pairs of neighbouring datasets so also datasets with very large values $R$.
> > >
> > > Thank you for your valuable question. Yes, we define $R$ as an upper bound for the entry value. It is meaningful in the following sense:
> > > - Even $K$ and $V$ are data dependent, we can calculate their maximum value before we add differential privacy noise, as $K$ and $V$ are known to the system.
> > > - The practical upper bound of $K$ and $V$ is small as many nomralization operations.
> > > - On the other hand, if we use float16, there is a natural upper bound $R=65,504$, see detail about [Half-precision floating-point format](https://en.wikipedia.org/wiki/Half-precision_floating-point_format).
> > > - Furthermore, given bounded input is quite a standard assumption for DP analysis due to sensitivity requirement, which is also mentioned by the reviewer.
> > >
> > > ### Q2: So this would lead to a DP $\varepsilon$ that scales with the sequence length? Using group privacy even moderate sequence lengths of 100 tokens or more significantly reduces the privacy protection.
> > >
> > > We thank the reviewer point out these questions. We admit that we are word/token level DP. However, our current utility additive error bound is $O(n^{-1})$ for one token (Line 23-24), which means that when the sequences become longer, each token has a smaller utility error under a fixed privacy constraint. When we consider the group privacy, say size $n$, we can initialize the data structure with $\epsilon/n$, and by Theorem 2.2 in [1], we have DP guarantee and additive error is independent with sequence/token length $n$. For relative error, it is independent of DP parameters and thus guaranteed.
> > >
> > > On the other hand, we would like to highlight our contribution that this work is the first work to provide differential privacy to cross-attention, as our title claimed. We make a step forward to provide differential privacy mechanisms guarrantee in transformers, one of the most important modules in deep learning. For sentence-level/document-level privacy, we agree that they are very important. We believe that our work is preliminary work to prepare for further analysis of more complicated practical tasks.
> > >
> > > [1] Dwork, C., & Roth, A. (2014). The algorithmic foundations of differential privacy. Foundations and Trends in Theoretical Computer Science.

---

> > > > ### Comment · Reviewer_s8oF · 2024-12-03
> > > >
> > > > Using the max representable number of float16 as the sensitivity requires noise of such large variance that I'm convinced would lead to a very poor utility and essentially random output of the mechanism. Furthermore, also the noise would be represented by floating point number which is an assumption not made in the proofs. I'm therefore hesitant to accept the suggestions as solutions to the sensitivity problem.
> > > >
> > > > Overall, while I appreciate the theoretical contributions of this paper, I still have doubts about its practical applicability. I am not convinced that this privacy mechanism provides a good enough privacy-utility trade-off required for typical applications of the cross-attention mechanism.
> > > >
> > > > I maintain my score as I think this paper can provide a stepping stone for the community to solve the problem but this paper does not provide a workable solution to privatize cross-attention

---

> > > > > ### Author Response · Authors · 2024-12-04
> > > > > **Thank you and final reply by authors**
> > > > >
> > > > > Thank you for recognizing the theoretical novelty of our work! We greatly appreciate your thoughtful feedback and address your concerns as follows:
> > > > >
> > > > > ### Q1: Sensitivity concerns
> > > > > We acknowledge your valid point regarding the large range of float16 values, which can result in a high sensitivity. However, our approach considers that $K$ and $V$ in the cross-attention mechanism have a natural upper bound $R$. This bound can be effectively reduced through various methods, such as incorporating an additional normalization layer within the attention module. We will further explore such strategies to refine the sensitivity.
> > > > >
> > > > > ### Q2: Practical applicability
> > > > > We understand your concerns about the privacy-utility trade-off. Our work provides a novel perspective on integrating DP into LLMs, along with a controlled and relatively small error bound. As this is an emerging topic, we view our contribution as a foundation for future development, and we are committed to exploring additional methods to enhance the practical utility of our approach.
> > > > >
> > > > > Thank you again for your thoughtful review. We value your feedback as it will help us and the community advance this important area of research.

---

### Official Review · Reviewer_Ypjs · 2024-11-10

**Soundness:** 3
**Presentation:** 3
**Contribution:** 3
**Rating:** 5
**Confidence:** 3

**Summary:**

This paper introduces a novel differential privacy (DP) mechanism, DPTREE, for preserving privacy in cross-attention mechanisms in large generative models (LGMs). The structure provides privacy guarantees while supporting adaptive queries with theoretically bounded errors and optimized memory and computation costs.

**Strengths:**

1.Novel Application of DP in Cross-Attention: The paper proposes the first DP solution specifically for cross-attention, a fundamental mechanism in LGMs, which is innovative and addresses privacy in an underexplored area.
2.heoretical Rigor: The paper is backed by solid theoretical proofs of privacy and accuracy guarantees, adding depth and credibility to the approach.
3.Adaptive Query Handling: The mechanism is robust against adaptive queries, making it suitable for real-world applications where inputs can vary significantly.
4.Memory and Computation Efficiency: Through the DPTREE structure, the proposed method optimizes memory usage and computational complexity, enhancing scalability.

**Weaknesses:**

1.Lack of Innovation in Core Techniques: The DPTREE mechanism heavily relies on polynomial kernel approximations and basic DP noise mechanisms, which are established techniques in DP literature. The limited novelty in these technical foundations weakens the originality of the proposed approach.
2.Limited Generalization and Practicality: DPTREE is specifically tailored to cross-attention, and its reliance on polynomial kernel approximations may not generalize well across other types of attention mechanisms or data structures in LGMs. This specificity reduces the model's adaptability, which could hinder broader applicability and limit its impact in the field.
3.Computational and Initialization Costs: Although optimized for memory during inference, the initialization and computational costs for high-dimensional data remain high. This could make the solution impractical for larger LGMs or applications requiring frequent real-time updates.

**Questions:**

1.Given the DPTREE's reliance on polynomial kernel approximations and its focus on cross-attention, how well does it generalize to other attention mechanisms or model architectures, especially in high-dimensional or heterogeneous data scenarios?
2.What strategies does the DPTREE mechanism have in place to manage potential approximation errors that may accumulate during adaptive querying?
3.Are there specific types of datasets or applications where DPTREE demonstrates significantly better or worse performance?

---

> ### Author Response · Authors · 2024-11-16
>
> We thank the reviewer for their valuable suggestions. We provide our response below and hope it will address your concerns.
>
> ### W1: Lack of Innovation in Core Techniques
> As indicated, the polynomial kernel method and DP noise techniques have been proposed before. However, we would like to emphasize that our analysis offers a new view to combine the two in a clear way.
> - We are the first work to reformulate the cross-attention computation to a weighted distance problem and then successfully apply those techniques. If we do not follow this approach, we cannot leverage the polynomial kernel method to approximate the $\mathsf{Softmax}$ function, which complicates the analysis of the DP guarantee. Additionally, we would lose the opportunity to utilize our simple yet efficient DPTREE data structures for solving the weighted distance problem. From our perspective, we believe our formulation and analysis are novel.
> - We also handle the adaptive query case that is critical in practice for an intentional attacker. We thank the reviewer’s appreciation of this contribution in the third point of the Strength section.
>
> ### W2: Limited Generalization and Practicality
> We thank the reviewer for pointing out this concern. We admit that our analysis is tailored to cross-attention due to technical hardness. To the best of the author’s knowledge, we are the first work to provide differential privacy for cross-attention.
>
> To extend to other settings, such as self-attention, we discussed this in Section 5, Line 479. In summary, the self-attention mechanism requires dynamic updates to tree nodes for each query, which are not supported in our current analysis. Our approach relies on separating the query and key-value matrices to pre-construct data structures. Applying this to self-attention necessitates dynamic updates, introducing challenges in DP analysis when supporting the update function in our DPTREE. However, with a carefully crafted and efficient design for handling dynamic updates, particularly in managing the varying length of the query matrix, these challenges can potentially be mitigated. With dynamic updates, it is possible to apply our current approach to self-attention. We leave this as future work.
>
> ### W3: Computational and Initialization Costs
> Our problem formulation abstracts scenarios where the key and value matrices are precomputed and stored in memory or disk, such as in RAG or system prompts (e.g., the latest example from Claude in this [link](https://www.anthropic.com/news/prompt-improver)). These pre-computed matrices are used to construct our data structure, so the inference time is not affected by the initialization time.
>
> ### Q1: How well does it generalize to other attention mechanisms or model architectures, especially in high-dimensional or heterogeneous data scenarios?
>
> We discuss the extension to self-attention in Section 5 in Line 479 and leave it as our future work. As for other model architectures, if the model can be reformulated into a weighted distance problem, our method can still apply. For the heterogeneous data scenarios, as our method did not depend on data distribution, we believe our methods can handle this situation.
>
> ### Q2: What strategies does the DPTREE mechanism have in place to manage potential approximation errors that may accumulate during adaptive querying?
>
> The error bound is guaranteed, as proved in Line 260, and is robust to adaptive queries, as demonstrated in Section H. While the error is controlled for individual queries, handling long chains of dependent queries falls outside our initial scope. Still, our current approach provides robustness to adaptive queries, which can provide the foundation for handling long chains of dependent queries. For example, our methods can ensure sentence-level or multi-token privacy by protecting each individual word—e.g., every token in a sentence can be perturbed. Designing a data structure specifically for sentence-level privacy protection presents an interesting and challenging problem. One potential approach is to leverage the concept of group privacy as outlined in [1], which allows neighboring datasets to differ by multiple tokens. We are open to further discussion and exploration of this topic based on the reviewer’s feedback.
>
> ### Q3: Are there specific types of datasets or applications where DPTREE demonstrates significantly better or worse performance?
>
> Our method does not assume data distribution and thus may hold for any dataset. For the application side, as the input context length increases, the error decreases. Our methods may perform better in long context datasets.
>
> [1] Dwork, C., & Roth, A. (2014). The algorithmic foundations of differential privacy. Foundations and Trends in Theoretical Computer Science.

---

> > ### Author Response · Authors · 2024-12-02
> > **Looking forward to receiving your feedback**
> >
> > Dear Reviewer Ypjs,
> >
> > We hope we have adequately addressed your issues. We would be very grateful if you could provide feedback on our rebuttal since the discussion deadline is approaching in one day. If you require further clarification or have any additional concerns, please do not hesitate to contact us. We are more than willing to continue communicating with you.
> >
> > Warmest regards,
> >
> > Authors

---

### Author Response · Authors · 2024-11-16
**Deepest thanks to all the thoughtful reviews**

We appreciate all reviewers' efforts and valuable comments. We are grateful that Reviewers Ypjs, s8of, 9PH2, and MwpB appreciate our novel DP cross-attention approach and the strong theoretical foundations of our method. They recognize the innovation in applying DP to cross-attention within LGMs, especially through the transformation of cross-attention into a “weighted distance problem,” which enables the use of established DP algorithms like the binary tree mechanism. Ypjs also highlights our adaptive query handling and the optimized efficiency of the DPTREE structure, which enhances scalability. Furthermore, Ypjs, s8of, 9PH2, and MwpB commend our detailed analysis of utility, computational complexity, and privacy properties, viewing it as a solid foundation for DP mechanisms in LGMs. MwpB specifically found Figure 1 in the appendix helpful for illustrating the connection between cross-attention and the DP binary tree algorithm.

Here we summarize all the updates (in color **blue**) we made in the revision. Note that all line numbers in the rebuttal correspond to the fourth version revision.
- Discuss the DP matrix mechanism in Section 5 Line 484-488
- Add more comparison with DP-SGD in Line 124-129
- Add DP distance related work in Section 4.2 in Line 353-358
- Add notations for O tilde in Line 152-155
- Add $KV$ private $Q$ public in Section 3 in Line 222-225

We will address each reviewer's questions and concerns in individual rebuttals.

---

### Author Response · Authors · 2024-11-24
**Further update in revision**

Per reviewer MwpB valuable suggestions, we made a second revision, where the update is in color **brown**. We summarize all major updates below. Note that all line numbers in the rebuttal correspond to the fourth revision version.
- Line 161-165: we redefine the neighboring dataset and sensitivity to the matrix version
- Line 189-208: we provide a new algorithm DPCrossAttention
- Line 228-232: we modify the statement of $D$
- Line 381-400: we modify our algorithm DPTreeSoftmax
- Line 2031-2137: we reanalyse the modified algorithm DPTreeSoftmax

---

### Author Response · Authors · 2024-11-26
**Third version revision update**

Per reviewer MwpB valuable suggestions, we made a third revision, where the update is in the color **purple** (we also keep our old updates). We summarize all major updates below. Note that all line numbers in the rebuttal correspond to the fourth revision version.
- Line 17-26: we modify abstract
- Line 84: we update our main theorem
- Line 245-267: we modify our main theorem and add one remark
- Line 1826-1833: we modify the paragraph of Section H.2
- Line 1870-1889: we elaborate the proof of Theorem H.8
- Line 2072-2137: we update the proof of our main theorem

---

### Author Response · Authors · 2024-11-28
**Fourth version revision update**

Per reviewer MwpB valuable suggestions, we made a fourth revision, where the update is in the color **green** (we also keep our old updates). We summarize all major updates below. Note that all line numbers in the rebuttal correspond to the fourth revision version.
We improve our DPTree to remove the relative error $\alpha$ previous introduce.
- Line 298-308: resign query algorithm
- Line 359-371: describe our new designed DPTree
- Line 1338-1364: provide new lemma to efficient calculate weighted $\ell_p^p$ distance
- Line 1337-1403: improve DPTreeDistance data structure
- Line 1418-1474: prove the privacy and utility of the improved DPTreeDistance
- Line 1881-1889: rewrite the proof of error analysis of DPTreeSoftmax
- Line 2088-2137: modify the proof of main theorem

---

### Meta-Review · Area_Chair_imRG · 2024-12-19

**Metareview:**

There were a lot of discussion on the paper. However, there were not many folks strongly supporting the work. Few of the concerns that lingered are:

a) Lack for experiments for a completely applied problem. It is unclear what the theorems convey without having real numbers on benchmark tasks.

b) The comparison to DP-SGD or standard DP training algorithms is limited. I think the only way to handle this issue is via having experiments with an end to end training task.

We would like to revise the paper focussing on these two aspects.

**Additional Comments On Reviewer Discussion:**

The rebuttal discussion did address few of the concerns. However, the lack of experiments lingered.

---

### Decision · Program_Chairs · 2025-01-22

Reject